

# Comparison of different ensemble assimilation methods in a modular hydrological model dedicated to water quality management

Emilie Rouzies [1], Claire Lauvernet [1], and Arthur Vidard [2]

[1]INRAE, RiverLy, Lyon-Villeurbanne, 69625 Villeurbanne Cedex, France
[2]Univ. Grenoble-Alpes, Inria, CNRS, Grenoble-INP, LJK, 38000 Grenoble, France

**Correspondence:** Claire Lauvernet (claire.lauvernet@inrae.fr)

**Abstract.** Hydrological models are valuable tools for understanding the movement of water and contaminants in agricultural catchments. They are particularly useful for assessing the impact of landscape organization on pesticide transfers and for developing effective mitigation strategies. However, using these models in an operational context requires reducing uncertainties in their outputs, which can be achieved through data assimilation methods. In this study, we aim to integrate surface moisture
images into the PESHMELBA water and pesticide transfer model using data assimilation techniques. Twin experiments are conducted on a virtual catchment consisting of vineyard plots and vegetative filter strips. We compare the performance of the Ensemble Kalman Filter (EnKF), the Ensemble Smoother with Multiple Data Assimilation (ES-MDA), and the iterative Ensemble Kalman Smoother (iEnKS) in jointly estimating vertical moisture profiles and some input parameters. Results indicate that the ES-MDA performs the best in estimating surface moisture and related input parameters, while all methods show
similar results for subsurface moisture variables and parameters. Furthermore, we examine the sensitivity of the methods to observation error magnitude, observation frequency, and ensemble size to establish an effective assimilation setup. This study paves the way for future operational applications of data assimilation in PESHMELBA.

## 1   Introduction

Understanding water and pesticide transfers is critically important to protect aquatic and human lifes. To do so, numerical sim-
ulations effectively support risk assessment studies at the catchment scale. These studies particularly benefit from distributed hydrological models as they can accurately describe the relevant processes and transfer pathways that influence pesticide fate.

Such hydrological models often simulate several physical processes in order to properly catch the complex reality of the field. They need large sets of input parameters, potentially varying in space, that may be hard to properly define. Uncertainties on parameters, so as on model structure can thus result in significant errors in model simulations.

Data assimilation (DA) is an interesting approach to quantify and reduce these uncertainties. It consists in combining complementary information from observations distributed in time and space and from a numerical model while accounting for uncertainties from both sources. Stochastic ensemble methods derived from the Kalman Filter (Kalman, 1960) such as the Ensemble Kalman Filter (EnKF, Evensen 2003, 2009) or the Ensemble Smoother (ES, van Leeuwen and Evensen 1996) are broadly used in geophysics. They consist in Monte Carlo algorithms and linear solutions of the estimation problem. Dimension





reduction is performed by approximating the state probability distribution by an ensemble of vectors. Such approximation makes these methods particularly suitable for high dimension problems (Katzfuss et al., 2016). Observations are integrated as they come, one at a time (filter) or by batches, in a temporal window (smoother). Although the EnKF and its variants are based on the assumption of Gaussian statistics, they have been successfully applied to nonlinear, non Gaussian problems in many fields over the last decades (e.g. Bertino et al., 2003; Crow and Wood, 2003; Moradkhani et al., 2005; Rochoux et al., 2014;

Kurtz et al., 2016; Devers et al., 2020).

In many of these studies, DA is used to jointly correct state variables and estimate model parameters. A classical approach to perform joint estimation consists in augmenting the state vector by including parameters to be estimated. Parameters are mostly non-observed and their posterior distributions are deduced from the covariances built to perform DA between them and the state variables (Bocquet and Sakov, 2013). In hydrology, wrong parameter values are often identified as the dominant

source of error (Hendricks Franssen and Kinzelbach, 2008; Nie et al., 2011) and joint estimation can significantly improve the simulation accuracy. Recent studies have demonstrated the potential of joint estimation in groundwater flow or integrated surface-subsurface hydrological models at the catchment or hillslope scale (e.g. Hendricks Franssen and Kinzelbach, 2008; Baatz et al., 2017; Botto et al., 2018). Among them, Pasetto et al. (2015) examines the potential of the EnKF in the integrated surface–subsurface hydrological model CATHY (Paniconi and Putti, 1994; Camporese et al., 2010) to estimate the field of sat-

urated hydraulic conductivity on a virtual hillslope. Xie and Zhang (2010) also demonstrate the EnKF capabilities to estimate the runoff curve number empirical parameter in addition to various prognostic variables in the conceptual hydrological SWAT model (Arnold et al., 1998). The ES capabilities are also investigated to perform joint estimation as this scheme is easier to implement and less computationally demanding than the EnKF. In Crestani et al. (2013), the performances of the EnKF and the ES are compared to deduce the spatial distribution of hydraulic conductivity using a groundwater flow and transport model.

Bailey and Baù (2012) also retrieve conductivity distributions based on the ES, both on its original version and on an iterative version. More recently, Emerick and Reynolds (2013a) introduce the Ensemble Smoother with Multiple Data Assimilation (ES-MDA), an iterative scheme based on the ES that allows for assimilating several times the same observations. This scheme has been shown to outperform both the EnKF and the ES for parameter estimation in reservoir history-matching problems (Emerick and Reynolds, 2013a, b). Cui et al. (2020) also use the ES-MDA to estimate soil hydraulic parameters from Hydrus-

1D simulations (Šimůnek et al., 1998) and water content observations. These studies focus solely on parameter estimation, but the ES-MDA scheme could easily be extended to joint-estimation in a hydrological modelling context.

Ensemble methods have also percolated into the variational DA community which defines the data assimilation problem by a cost function to be minimized. Ensemble variational methods such as the iEnKS (iterative Ensemble Kalman Smoother, Bocquet and Sakov 2013) have recently been developed and have also demonstrated their abilities for solving joint estimation

problems. The iEnKS for joint estimation consists in iteratively minimizing a cost function that depends on the augmented state. Contrarily to classical variational problems, the cost function is defined in the ensemble space which limits the required computational effort. This DA method has been tested on simple oceanic models (Bocquet and Sakov, 2013) and on atmo-sphere chemistry models (Defforge et al., 2018) so far and has been shown to outperform the EnKF in some cases. In Bocquet and Sakov (2013), the authors argue that this method may be able to successfully deal with non linearities. The iEnKS has





not yet been applied to hydrological models but it may be worth exploring to deal with the complex structures and physical processes implemented in such models.

The above references show that there have been a wealth of studies dedicated to DA in catchment-scale hydrological models, both conceptual and physically-based. However, during the last decade, a novel approach for modelling has been emerging in hydrology. This approach consists in building physically-based and distributed models with a modular structure, relying

for example on flexible modelling frameworks (Buytaert et al., 2008). Resulting models are then composed of distinct code units standing for various physical processes that are coupled in the modelling framework. Motivations behind this innovative philosophy is to provide model structures that are flexible enough to evolve according to the targeted application. Several tools already exist in the hydrological field and they show promising results for risk assessment applications (Tortrat, 2005; Moussa et al., 2010; Branger et al., 2010; Kraft et al., 2012; Rouzies et al., 2019). However, these modular models are often charac-

terised by a highly interactive structure and numerous parameters, which complicates uncertainty quantification and reduction (Rouzies et al., 2023). As a result, the application of DA to these models may lead to different results from those of classical ones, warranting an in-depth study and comparison of the available DA algorithms.

Within this context, the objectives of this study are twofold. First, we aim at proposing first examples of a DA framework in a process-oriented, modular model used for risk assessment applications. Second, we propose to compare three DA algorithms

(namely the EnKF, the iEnKS and the ES-MDA) that are representative of the 3 main types of DA ensemble methods: a filter, a hybrid variational/ensemble smoother that is efficient over short data assimilation windows and an ensemble smoother that is efficient over long data assimilation windows. To that effect, we apply these DA schemes to the PESHMELBA model (PESticide and Hydrology Modelling at the catchment scale, Rouzies et al. 2019). PESHMELBA aims to simulate different landscape configuration scenarios and rank them in terms of pesticide transfer mitigation. Such model could then highly benefit

from joint estimation as it could contribute to identify consistent parameter distributions for scenario exploration. In this paper, we focus on the joint estimation of moisture variables and input parameters $\theta_{sat}$ (water content at saturation) for different soil types by assimilating surface soil moisture images. We do so on the basis of synthetic experiments and we compare the EnKF, the ES-MDA and the iEnKS performances. The outline of the paper is as follows: after a short description of the PESHMELBA model in Section 2.1, the different DA methods are introduced in Section 2.2. The case study and the DA setup are presented in

Section 2.3 and Section 2.4. The capabilities of the tested methods to retrieve variables and input parameters for both surface and subsurface compartments are explored in Section 3.1. Sensitivities to error prescription, ensemble size and observation frequency are investigated in Section 3.2.

## 2 Material and methods

### 2.1 Model description

The PESHMELBA model is a physically-based and spatialized model that simulates water and pesticide transfers at the scale of small agricultural catchments (Rouzies et al., 2019). It is designed to compare different landscape management scenarios and to identify an optimal configuration regarding pesticide transfer mitigation. As PESHMELBA aims at assessing the impact





of landscape composition, the hydrodynamical impact of various landscape elements such as Vegetative Filter Strips (VFSs), hedges, or ditches is explicitly simulated in model units that interact ones with others. PESHMELBA meshing is based on the landscape organisation as one mesh element stands for one landscape feature. As a result, physical process representation and coupling are adapted to this heterogeneous, element-scale meshing.

In this study, we use a version of PESHMELBA that simulates surface/subsurface hydrological fluxes within vineyard plots, VFSs and river reaches. The discretization for each landscape element and the physical processes integrated in this code version are summarized in Figure 1. Each plot or VFS is simulated as a single column of soil discretised vertically into numerical cells of heterogeneous thickness. Each soil column is composed of one or several soil layers, also called soil horizons, that are characterised by distinct hydrodynamical behaviours. Within a time step, vertical infiltration is solved on each column in the catchment, using the 1D-Richards equation for flows in variably saturated porous media (Richards, 1931). Once vertical infiltration has been solved, the overland flow is sequentially computed from ponding height resulting from Richards equation solving. Overland flow toward downstream elements is simulated using a pseudo-1D kinematic wave (Lighthill and Whitham, 1955). Channel flow routing in the river is also simulated using a kinematic wave approximation in a network of reaches with trapezoidal sections. For both overland and channel flows, a refined time step is used compared with vertical infiltration calculation. Finally, at the end of the time step, subsurface lateral flows in the saturated zone between plots or VFSs are calculated based on Darcy law (Darcy, 1857). Lateral saturated exchanges between water tables and the river are computed from Miles equation (Miles, 1985) adapted by Dehotin (2007). As a result, PESHMELBA is able to simulate the hydrological variables and their dynamics in 3 dimensions.

All physical processes are integrated in PESHMELBA in independent code units. The different code units are coupled in the OpenPALM coupler (Fouilloux and Piacentini, 1999; Buis et al., 2006) that ensures variable exchanges and synchronization. It is worth emphasising that PESHMELBA final structure is characterised by strong interactions, non linearities and threshold effects. It makes any attempt of DA particularly challenging and justifies studying several approaches in depth in order to identify the most appropriate one.





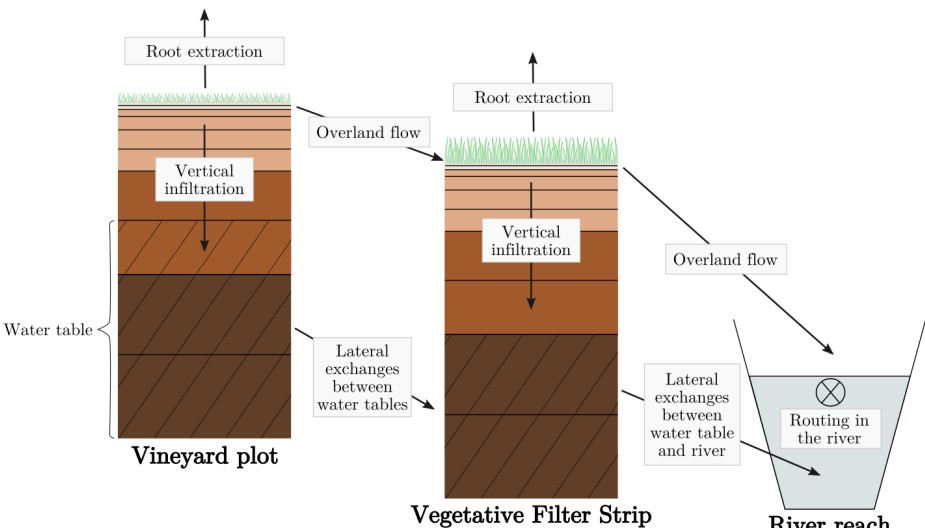

**Figure 1.** Discretisation of landscape elements and physical processes included in the PESHMELBA version of this study. The horizontal black lines in the left and middle soil columns delimit the numerical cells, while the hatched areas represent the water tables. In these columns, the different brown fillings delimit the different soil horizons.

## 2.2 Data assimilation methods

The following sections introduce the ensemble Kalman filter, the ensemble smoother with multiple data assimilation and the iterative ensemble Kalman smoother with multiple data assimilation. These methods aim to estimate the probability density function (pdf) of a state vector conditionally to available observations. In this case study, we aim to estimate 3D-moisture profiles and some input parameters of the PESHMELBA model. The corresponding system state vector at time $t_k$ with $k \in \{1, ..., K\}$ is denoted $\mathbf{x}_k \in \mathbb{R}^n$ in what follows. We assume available observations of mean moisture in the top 5cm of the soil and the corresponding observation vector at time $t_k$ is denoted $\mathbf{y}_k^o \in \mathbb{R}^p$.

The model $\mathcal{M}_{k \rightarrow k+1} : \mathbb{R}^n \rightarrow \mathbb{R}^n$ propagates the state vector from time $t_k$ to time $t_{k+1}$ (Eq. 1) and $\mathcal{H}_k : \mathbb{R}^n \rightarrow \mathbb{R}^p$ is the observation operator that maps the state variable space onto the observation space at each time step (Eq. 2):

$$\mathbf{x}_{k+1} = \mathcal{M}_{k \rightarrow k+1}(\mathbf{x}_k) + \mathbf{v}_k, \tag{1}$$

$$\mathbf{y}_k^o = \mathcal{H}_k(\mathbf{x}_k) + \mathbf{z}_k, \tag{2}$$

where $\mathbf{v}_k$ and $\mathbf{z}_k$ are the state error and the observation error respectively. They are assumed to be unbiased and to follow a Gaussian distribution:

$$\mathbf{v}_k \sim \mathcal{N}(\mathbf{0}, \mathbf{P}_k), \tag{3}$$

$$\mathbf{z}_k \sim \mathcal{N}(\mathbf{0}, \mathbf{R}_k), \tag{4}$$

where $\mathbf{P}_k \in \mathbb{R}^{n \times n}$ (resp. $\mathbf{R}_k \in \mathbb{R}^{p \times p}$) is the model error covariance matrix (resp. observation error covariance matrix).

In the case of joint estimation, the state vector $\mathbf{x}_k$ is augmented with model parameters to be estimated and the model $\mathcal{M}_{k \rightarrow k+1}$





also includes an evolution law for the estimated parameters in addition to the state dynamical evolution. As soil characteristics are not expected to change over time at the scale of interest, we have chosen a persistence law to represent the (non-)evolution

of the parameters.

The DA methods introduced in the next sections are sequential and consist of an alternation of a *forecast* step and an *analysis* step. During the forecast step, the model propagates a background state $\mathbf{x}_{k-1}$ from $t_{k-1}$ to $t_k$ resulting in a forecast state vector $\mathbf{x}_k^f$. This prior state is updated during the analysis step based on the available measurements. The analysis results in a posterior state $\mathbf{x}_k^a$ at time $t_k$ that becomes the background for the next forecast step between $t_k$ and $t_{k+1}$. Accordingly, any given vector

$\mathbf{a}$ or matrix $\mathbf{A}$ derived from the forecast step is denoted $\mathbf{a}^f$ or $\mathbf{A}^f$ in what follows while it is denoted $\mathbf{a}^a$ or $\mathbf{A}^a$ if it is calculated at the analysis step. Although they aim to solve the same problem derived from the Bayes theorem, the methods tested in this study differ in the time window that is considered in the forecast step and the number of measurements simultaneously used in the analysis step. The distinct approximations and assumptions they are based on may lead to significantly different solutions that will be analyzed in this study. Their common structure and characteristics are summarized in Figure 2.

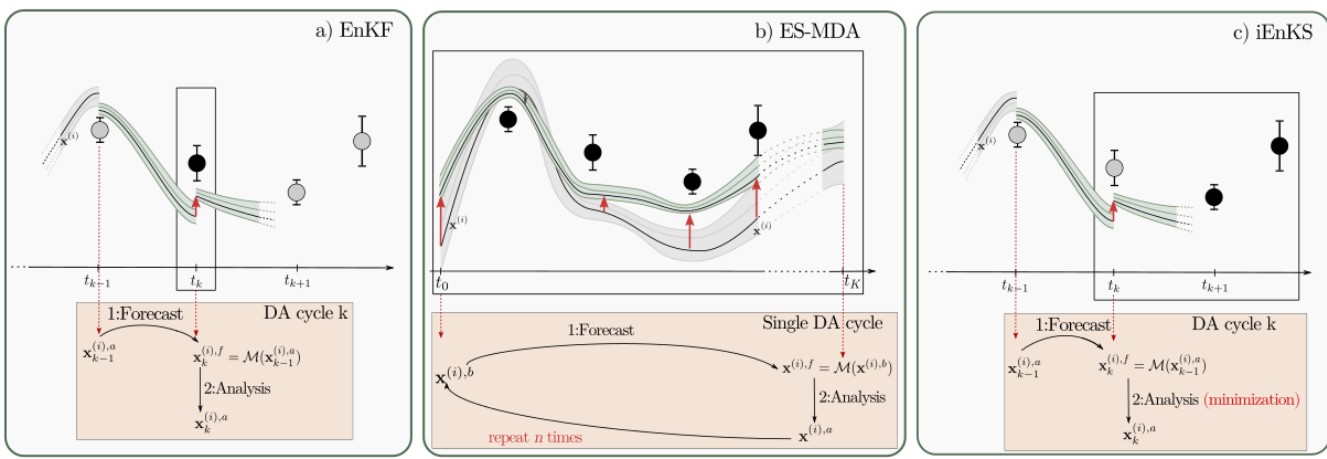

**Figure 2.** Schematic view of one DA cycle for the EnKF (left), the ES-MDA (middle) and the iEnKS (right). Lines depict trajectories for some members of the ensemble and the coloured envelop stands for the ensemble uncertainty. Dots depict the available observations. The black dots in the black window are the observations that are used for the current DA cycle whereas grey dots are non-used observations.

**2.2.1 Ensemble Kalman Filter**

The Ensemble Kalman Filter (EnKF, Evensen 1994) extends the Kalman Filter resolution of the Bayesian estimation problem to non linear, high dimensional contexts. The state distribution is approximated by an ensemble of $M$ state vectors $\mathbf{x}^{(i)}, i \in \{1, ..., M\}$. Each vector is sequentially propagated during the forecast step by applying the nonlinear model $\mathcal{M}$ and updated during the analysis step using the current observations (see Figure 2, left). This study is based on the Ensemble Transform

Kalman Filter (ETKF, Bishop et al. 2001; Hunt et al. 2007). At each time $t_k$, we consider $\mathbf{X}_k^f \in \mathbb{R}^{n \times M}$, the matrix of normalized





perturbations whose columns are expressed as:

$$[\mathbf{X}_k^f]_i = \frac{\mathbf{x}_k^{(i),f} - \overline{\mathbf{x}}_k^f}{\sqrt{M-1}}. \tag{5}$$

with

$$\overline{\mathbf{x}}_k^f = \frac{1}{M} \sum_{i=1}^{M} \mathbf{x}_k^{(i),f}, \tag{6}$$

During the analysis step, one looks for a state vector $\mathbf{x}_k^a$ in the affine subspace spanned by the anomalies: $\overline{\mathbf{x}}_k^f + \mathrm{Vec}\{\mathbf{x}_k^{(1),f} - \overline{\mathbf{x}}_k^f, ..., \mathbf{x}_k^{(M),f} - \overline{\mathbf{x}}_k^f\}$. The analysis is then expressed as:

$$\mathbf{x}_k^a = \overline{\mathbf{x}}_k^f + \mathbf{X}_k^f \mathbf{w}_k \tag{7}$$

where $\mathbf{w}_k \in \mathbb{R}^M$ is a weight vector in the ensemble subspace. The optimal weight vector is obtained from the Kalman Filter equation:

$$\mathbf{x}_k^a = \overline{\mathbf{x}}_k^f + \mathbf{K}_k [\mathbf{y}_k^o - \mathcal{H}_k(\overline{\mathbf{x}}_k^f)], \tag{8}$$

where the Kalman gain $\mathbf{K}_k = \mathbf{P}_k^f \mathbf{H}_k^\top (\mathbf{H}_k \mathbf{P}_k^f \mathbf{H}_k^\top + \mathbf{R}_k)^{-1}$ is computed with the forecast error covariance matrix expressed from the ensemble: $\mathbf{P}_k^f = \mathbf{X}_k^f \mathbf{X}_k^{f^\top}$. Identification of terms using Sherman-Morrison-Woodbury formula allows to express the optimal weight vector $\mathbf{w}_k$ in the ensemble subspace:

$$\mathbf{w}_k = (\mathbf{I}_M + \mathbf{Y}_k^{f^\top} \mathbf{R}_k^{-1} \mathbf{Y}_k^f)^{-1} \mathbf{Y}_k^{f^\top} \mathbf{R}_k^{-1} \delta_k, \tag{9}$$

where $\delta$ is the innovation vector $\delta_k = \mathbf{y}_k^o - \mathcal{H}_k(\overline{\mathbf{x}}_k^f)$ that contains the observations and where $\mathbf{Y}_k^f \in \mathbb{R}^{p \times M}$ contains the observation normalized perturbations:

$$[\mathbf{Y}_k^f]_i = \frac{\mathcal{H}_k(\mathbf{x}_k^{(i),f}) - \overline{\mathbf{y}}_k^f}{\sqrt{M-1}}, \tag{10}$$

with:

$$\overline{\mathbf{y}}_k^f = \frac{1}{M} \sum_{i=1}^{M} \mathcal{H}_k(\mathbf{x}_k^{(i),f}). \tag{11}$$

The posterior ensemble of perturbations is generated so as to be representative of the posterior uncertainty that can be factorized as $\mathbf{P}_k^a = \mathbf{X}_k^a \mathbf{X}_k^{a^T}$. The analysis normalized anomalies are then derived:

$$\mathbf{X}_k^a = \mathbf{X}_k^f (\mathbf{I}_M + \mathbf{Y}_k^{f^\top} \mathbf{R}^{-1} \mathbf{Y}_k^f)^{-\frac{1}{2}}, \tag{12}$$

This leads to the following expression for the analysed members $\mathbf{x}_k^{(i),a}$, $\forall i \in \{1, ..., M\}$:

$$\mathbf{x}_k^{(i),a} = \overline{\mathbf{x}}_k^f + \mathbf{X}_k^f (\mathbf{w}_k + \sqrt{M-1}[(\mathbf{I}_M + \mathbf{Y}_k^{f^\top} \mathbf{R}^{-1} \mathbf{Y}_k^{f^\top})^{\frac{-1}{2}}]_i), \tag{13}$$

The ETKF is favoured for high dimension problems as it alleviates the computational cost of the analysis. Indeed, most of the algebraic calculations are performed in the ensemble subspace which is generally of smaller dimension than the state or the observation space.





### 2.2.2 Ensemble Smoother with Multiple Data Assimilation

The above section has introduced a filtering approach that is a sequential scheme based on incremental updates of the state
vector $\mathbf{x}$. For each update, the present system state pdf is corrected by Eq. 12 using present observations only. In contrast, the
Ensemble Smoother (ES, van Leeuwen and Evensen 1996) aims to estimate a posterior distribution for the state vector in an
assimilation window $[t_1, ..., t_K]$ relying on all observations available in that window. First, the ensemble is integrated over the
whole assimilation window in a single forecast step. The augmented state vector $\mathbf{x}$ is composed of temporal trajectories for
each state variable, at each point of the model grid plus input parameters. The same analysis as for EnKF is then carried out,
using all the observations at the same time (see Figure 2, middle). The state variable trajectories are updated in space and time
through space-time covariances estimated from the ensemble (Crestani et al., 2013).

More recently, Emerick and Reynolds (2013a) introduced the Ensemble Smoother with Multiple Data Assimilation (ES-
MDA), an "iterative-version" that consists in iteratively performing the ES-like sequence: ensemble model integration over
the whole assimilation window followed by an analysis step. To be noted that the number of iterations is prescribed before
launching the DA procedure and does not depend on a convergence criterium. At iteration $(j)$, the analysis step from Eq. 12 is
replaced by:

$$\mathbf{X}^a_{(j)} = \mathbf{X}^f_{(j)}(\mathbf{I}_M + \mathbf{Y}^{f^\top}\alpha^{-1}_{(j)}\mathbf{R}^{-1}\mathbf{Y}^f)^{-\frac{1}{2}}, \tag{14}$$

where the notation $\mathbf{X}^a_k$ (resp. $\mathbf{X}^f_k$ and $\mathbf{Y}^f_k$) for time $t_k$ has been simplified to $\mathbf{X}^a$ (resp. $\mathbf{X}^f$ and $\mathbf{Y}^f$) and where the weight
$\alpha_{(j)}$ is an inflation term for the observation error covariance matrice $\mathbf{R}$. Considering $J$, the predefined total number of iterations
in the ES-MDA full scheme, the weights $\alpha_{(j)}, j \in \{1, ..., J\}$ must satisfy:

$$\sum_{j=1}^{J} \frac{1}{\alpha_{(j)}} = 1. \tag{15}$$

In this study, we set $\alpha_{(j)} = J, \forall j \in \{1, ..., J\}$ (Emerick and Reynolds, 2013a; Cui et al., 2020).

The model integration at iteration $(j+1)$ is then initialized with the posterior distribution of parameters obtained from the
analysis at iteration $(j)$. This scheme replaces a single abrupt analysis by several smaller analysis.

### 2.2.3 Iterative Ensemble Kalman Smoother with Multiple Data Assimilation

As with the EnKF, the iEnKS alternates forecast and analysis steps to perform incremental updates of the state. However, in
this fixed-lag smoothing context, each analysis aims to update a state vector at time $t_k$ using observations between $t_{k+1}$ and
$t_{k+L}$ where $L$ is the length of the Data Assimilation Window (DAW, see Figure 2, right). Moreover, contrarily to the EnKF
or the ES-MDA, the iEnKS fundamentally belongs to the category of variational methods. As such, the iEnKS analysis step
consists in minimizing a cost function deduced from the Bayesian estimation problem formulation:

$$\mathcal{I}(\mathbf{x}^a_k) = \frac{1}{2}\|\mathbf{x}^a_k - \mathbf{x}^f_k\|^2_{\mathbf{P}^f_k} + \sum_{l=1}^{L}\frac{1}{2}\|\mathbf{y}_{k+l} - \mathcal{H}_{k+l} \circ \mathcal{M}_{k \to k+l}(\mathbf{x}^a_k)\|^2_{\mathbf{R}_{k+l}}, \tag{16}$$





where $\|\mathbf{x}\|_{\mathbf{A}}^2 = \mathbf{x}^\top \mathbf{A}\mathbf{x}$. Like the EnKF or the ES, the iEnKS is an ensemble-based method and the calculations rely on $M$ members: $\mathbf{x}^{(i)}, i \in \{1,...,M\}$. $\mathbf{X}$ denotes the perturbation matrix (see Eq. 5).

At time $t_k$, the analysed state vector is expressed as an incremental correction $\mathbf{x}_k^a = \overline{\mathbf{x}}_k^f + \mathbf{X}_k^f \mathbf{w}$ and the minimization consists in finding the weight vector $\mathbf{w}^* \in \mathbb{R}^M$ that minimizes the cost function expressed in the ensemble subspace. Again, the forecast error covariance matrix is expressed from the ensemble: $\mathbf{P}_k^f = \mathbf{X}_k^f {\mathbf{X}_k^f}^\top$ leading to the following expression for the cost function:

$$\mathcal{I}(\mathbf{w}) = \frac{1}{2}\|\mathbf{w}\|^2 + \sum_{l=1}^{L} \frac{1}{2}\|\mathbf{y}_{k+l} - \mathcal{H}_{k+l} \circ \mathcal{M}_{k\to k+l}(\overline{\mathbf{x}}_k^f + \mathbf{X}_k^f \mathbf{w})\|_{\mathbf{R}_{k+l}}^2, \tag{17}$$

where $\|\mathbf{w}\|^2 = \mathbf{w}^\mathrm{T}\mathbf{w}$.

The minimization of the cost function $\mathcal{I}$ is performed in the ensemble subspace by a Gauss-Newton algorithm:

$$\mathbf{w}_{(j+1)} = \mathbf{w}_{(j)} - \mathrm{H}_{(j)}^{-1} \nabla\mathcal{I}_{(j)}(\mathbf{w}_{(j)}), \tag{18}$$

where $(j)$ is the iteration index, $\mathrm{H}$ is an approximation of the Hessian of $\mathcal{I}$ and $\nabla$ is the gradient operator. Such minimization actually corresponds to a nonlinear update.

The approximated Hessian and the gradient are expressed using the tangent linear of the operator transporting from the ensemble space to the observation space $\mathcal{H} \circ \mathcal{M}$: $\mathbf{Y}_{k+l,(j)} = [\mathcal{H}_{k+l} \circ \mathcal{M}_{k\to k+l}]'_{|\mathbf{x}_{k,(j)}} \mathbf{X}_k^f$. In this study and as proposed in Bocquet and Sakov (2013), this tangent linear operator is estimated using finite differences ("bundle" iEnKS version). Both $\mathrm{H}$ and $\nabla\mathcal{I}$ are then computed from the ensemble:

$$\mathbf{x}_{k,(j)}^a = \overline{\mathbf{x}}_k^f + \mathbf{X}_k^f \mathbf{w}_{(j)}, \tag{19}$$

$$\nabla\mathcal{I}_{(j)} = \mathbf{w}_{(j)} - \sum_{l=1}^{L} \alpha_l \mathbf{Y}_{k+l,(j)}^T \mathbf{R}_{k+l}^{-1}[\mathbf{y}_{k+l} - \mathcal{H}_{k+l} \circ \mathcal{M}_{k\to k+l}(\mathbf{x}_{k,(j)}^a)], \tag{20}$$

$$\mathrm{H}_{(j)} = \mathbf{I}_M + \sum_{l=1}^{L} \alpha_l \mathbf{Y}_{k+l,(j)}^T \mathbf{R}_{k+l}^{-1} \mathbf{Y}_{k+l,(j)}, \tag{21}$$

where $\alpha_l, l \in \{1,...,L\}$ are inflation weights for the observation error. As the ensemble size is generally rather small, $\mathrm{H}_{(j)}$ can be inverted using direct exact methods. As with the ES-MDA, the iEnKS implementation chosen in this paper allows each observation to be used several times. The weights $\alpha_l$ then merely measure the impact of each observation in the DAW. The sum of weights for an assimilation cycle respects:

$$\sum_{l=1}^{L} \alpha_l = 1 \tag{22}$$

In Bocquet and Sakov (2014), a heuristic justification of this MDA scheme is given. The reader should refer to this paper for more details about this scheme. In this study, we choose a uniform scheme so that $\alpha_l = \frac{1}{L}, \forall l = 1,..,L$.

The minimization takes place until a convergence criteria is reached. Then, the analyzed members $\mathbf{x}_k^{(i),a}, \forall i \in \{1,...,M\}$ are deduced at time $t_k$ using the mimimized vector $\mathbf{w}^*$ and the associated state vector $\mathbf{x}_k^a$:

$$\mathbf{x}_k^{(i),a} = \mathbf{x}_k^a + \sqrt{M-1}[\mathbf{X}_k^f \mathrm{H}^{-\frac{1}{2}}]_i, \tag{23}$$





This way, the iEnKS offers to perform the non linear update of the state as in standard variational DA (Carrassi et al., 2018).

## 2.3 Case study

In this study, we aim to investigate the accuracy and the robustness of the DA methods aforementioned in a simulation exercise based on PESHMELBA. To do so, we perform synthetic (or twin) DA experiments on a simplified, virtual catchment. DA experiments are conducted on the Morcille-like virtual catchment described in Rouzies et al. (2023). This size-limited catchment

is inspired from La Morcille catchment, situated in the Beaujolais vineyard region, east of France. A comprehensive description of the setup can be found in such study and its main characteristics are reported in what follows. The virtual catchment is composed of 10 vineyard plots, 4 VFSs and a portion of river that delimits two hillslopes. All soil columns are 4-m deep. They are vertically discretized in 25 numerical cells whose thickness ranges from 0.05m at the top to 1m at the bottom. Three soil units (SU), mainly sandy, compose the catchment in accordance with the soil composition of La Morcille catchment. Each soil

type is made up of the vertical succession of 3 or 4 soil horizons. Each soil horizon is characterised by its own hydrodynamical behaviour that results in distinct sets of parameters. Figure 3 summarizes the scenario composition, the different SUs and their vertical constitution. Vineyard plots and VFSs from the same SU are parametrized identically except for the first soil horizon. The first soil horizon is chosen to extend to the first 15cm in VFSs while it can vary on vineyard plots. Its main parameters are set in order to account for the effect of increased soil structuring and dense vegetation that mitigate surface runoff on VFSs.

This way, the saturated hydraulic conductivity, the ponding height and the roughness are increased compared with vineyard plots.

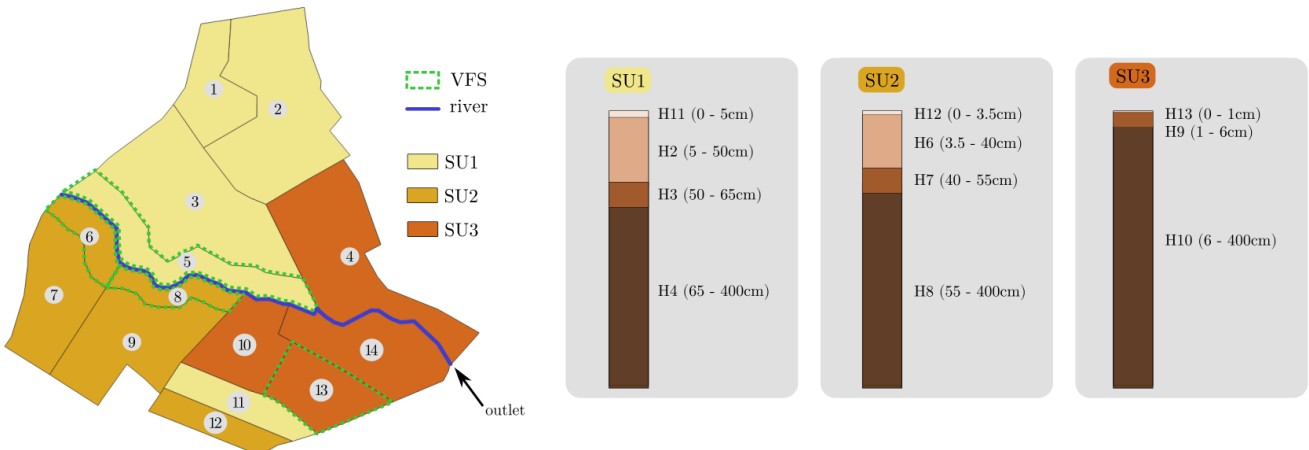

**Figure 3.** Left: composition and spatial distribution of SUs in the virtual catchment. VFSs are bounded with green dashed line while remaining mesh units are vineyard plots. Right: Details of SU constitution in terms of succession of soil horizons on vineyard plots. SU constitutions for VFSs are the same except for the surface horizon that is 15-cm deep.

Vineyard plot and VFS pressure profiles are initialized considering hydrostatic equilibrium and initial water table levels that are consistent with field data for the given time period. Realistic rain and potential evapotranspiration forcings that correspond





to a typical winter period in the area are used. Except for atmospheric forcings, all boundary conditions are zero-fluxes. Two vegetation types are set in this scenario. A vineyard cover is considered on plots while a permanent, mature, grassland is parametrized on VFSs. Root depth and root density are supposed to be constant all over the simulation for both of them while the Leaf Area Index (LAI) evolves over the simulation period for vineyard cover. LAI remains constant on grassland.

The full scenario results in 128 parameters whose nominal values and meaning are described in Appendix A. Some in-
put parameters may seem redundant from one soil horizon to another but we explicitly distinguish several horizons in order to prepare setups that include pesticide transfers. In such application, parameters will need to vary from one horizon to the other.

As a reference for result interpretation, Figure 4 shows an example of nominal time series of moisture at different depths (surface, intermediate and deep). At surface, soil moisture dynamics is strongly driven by rain forcings all over the simulation.
On the contrary, moisture varies significantly only after 1,200h of simulation at 0.2m depth, when stronger rain events occur. Finally, at 4m depth the system is not affected by the dynamics from superficial cells nore climate forcings. It remains in a steady state that corresponds to the presence of a water table, leading to a constant moisture value.

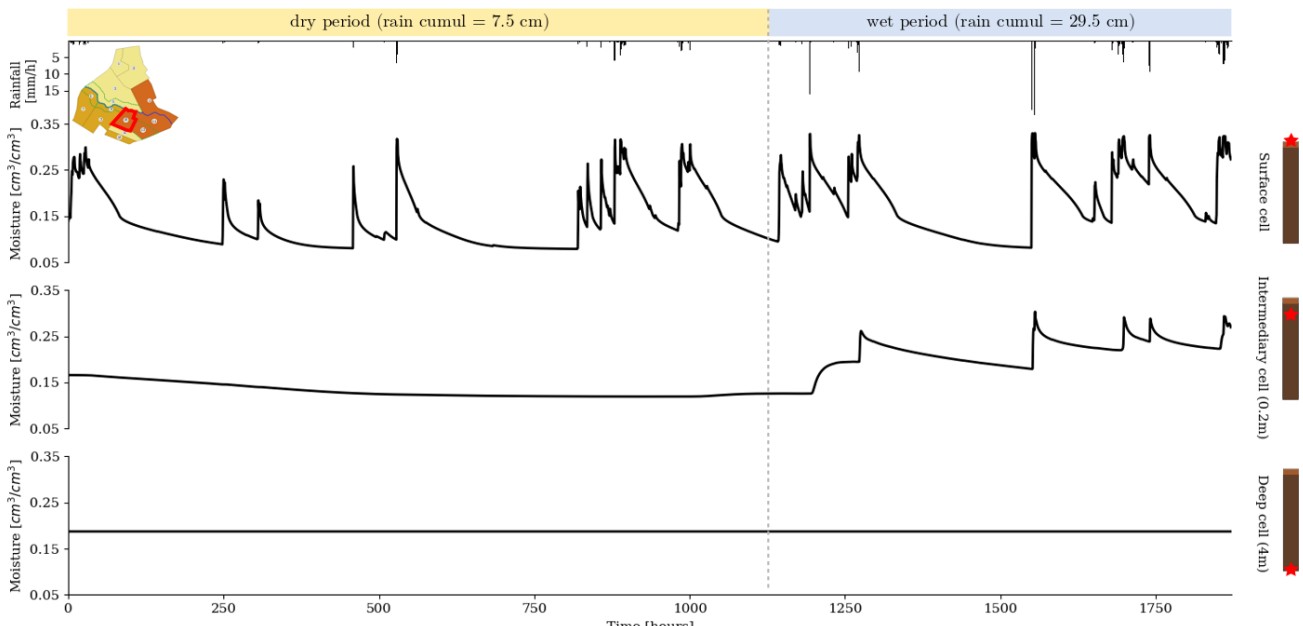

**Figure 4.** Soil moisture time series for the nominal simulation on plot 10 at surface (top), 0.2m depth (middle) and 4m depth (bottom). Rain forcings are shown in the top black histogram while the position of plot 10 in the catchment is denoted by a red line on the top-left pictogram. The soil composition of plot 10 is reminded in the soil columns depicted on the right with red stars showing the corresponding depths.

## 2.4 DA setup

As this study focuses on twin experiments, we use synthetic images that mimick satellite moisture data in the top 5 cm.
Such data are supposed to come from the synergistic use of optical and radar signals from Sentinel-1 and Sentinel-2 satellites





(Bousbih et al., 2018) which in turn can be converted into maps of surface soil moisture in agricultural areas (El Hajj et al., 2017; Gao et al., 2017). The synthetic images are generated from a PESHMELBA "True" reference run and perturbed with Gaussian, non biased noise. Observation errors are supposed to be uncorrelated in time and space and the standard deviation for error is set to 0.02 cm$^3$cm$^{-3}$ at first step. To be noted that this approximate for observation error is underestimated compared with

values found in Bousbih et al. (2018). In this paper, the authors conducted experiments on cereal crops. To our knowledge, similar experiments on vineyard have not been performed yet. Observation error for such soil cover may highly differ and sensitivity to observation error magnitude is thus investigated in this study. We assume available a time series of mean moisture in the top 5 cm for each vineyard plot and VFS in the catchment. The observation operator $\mathcal{H}$ is accordingly built as a matrix so as to relate each observation to the weighted mean of moisture over the 6 numerical cells that compose the top 5 cm of the

soil column:

$$\theta_{5cm} = \frac{\sum_{j=1}^{6} \theta_j dz_j}{\sum_{j=1}^{6} dz_j}, \tag{24}$$

where $\theta_j$ and $dz_j$ are the soil moisture and the thickness of numerical cell $j$ respectively.

For the sake of simplicity, we consider that the input parameters are the only sources of uncertainty in the model. The ensemble is initialized by independently perturbing the 128 input parameters. Perturbations are set according to each input parameter

pdf. Such pdf have been defined from field measurements, literature or expert knowledge in a previous study (Rouzies et al., 2023). They are reminded in Appendix A.

Joint estimation is performed in order to estimate both vertical moisture profiles and relevant uncertain input parameters. The global sensitivity analysis of PESHMELBA in this case study showed that parameters that most influence moisture profiles are mainly $\theta_s$ (water contents at saturation) for the different soil horizons (Radišić et al., 2023). The augmented state vector thus

includes such parameters for both surface and deeper soil horizons and a bias is added to their pdfs when generating the initial ensemble.

For the iEnKS, the number of Gauss-Newton iterations in the cost function minimization is set to 3 which, from experience, can limit the cost and be considered enough (Eq 2.2.3 being solved exactly). A preliminary sensitivity study also allowed to set the iEnKS optimal DAW length $L$ so as to assimilate observations by batch of 5. Similarly, from preliminary trials, the number

of iterations is set to 3 for the ES-MDA.

The nominal scenario consists in a 78-days simulation with 6-days observation frequency, standard deviation for observation error equal to 0.02 cm$^3$cm$^{-3}$ and ensemble size of 50 members. Then, a serie of experiments is set up in order to assess the robustness of the DA methods and to test their sensitivity to 1) observation error magnitude 2) observation frequency and 3) ensemble size. To do so, each factor (observation error magnitude, observation frequency and ensemble size) is varied

individually while the others are set up to nominal values. Tested values are reported in Table 1.





| observation error sd [cm³cm⁻³] | 0.001 **0.020** 0.040 0.060 0.080 0.100 0.120 0.140 0.160 0.180 0.200 0.300 0.400 |
|---|---|
| observation frequency [days] | 1 2 3 4 5 **6** 8 9 10 |
| ensemble size [-] | 20 **50** 100 200 500 |

**Table 1.** Explored scenarios to assess the sensitivity of DA methods to (i) observation error (standard deviation), (ii) frequency of observation and (iii) ensemble size. Nominal values are in bold.

The performances of the DA methods in the different setups are assessed by computing the Continuous Ranked Probability Score (CRPS, Brown 1974) on moisture time series, on each vineyard plot and VFS at surface and at 0.2 m and 4 m depths. The CRPS expresses the distance between the cumulative density function (cdf) of the probabilistic forecast and the cdf of a reference. In this case, the reference is the PESHMELBA "True" run that is supposed to be free of error. Its cdf is built using the Heaviside function and the CRPS consists in building the quadratic distance between the two functions:

$$CRPS(Y,\hat{\theta}) = \int_{-\infty}^{\infty} [F_Y(s) - H(s - \hat{\theta})]^2 ds, \tag{25}$$

where $F_Y$ is the cdf of the one-dimensional ensemble of moisture values Y, $\hat{\theta}$ is the deterministic value of moisture from the PESHMELBA "True" run and $H(x)$ is the Heaviside function. It is a positive error criteria variable: the closer to 0, the better the ensemble forecast is. The CRPS is expressed in the same unit as the evaluated variable. It is averaged over time/space in case of a multidimensional forecast $\mathbf{Y}$. In this study, CRPS scores are estimated following Hersbach (2000) and the corresponding formulation is detailed in Appendix B. In order to better capture the impact of DA on the simulation, we also use the Continuous Ranked Probability Skill Score (CRPSS), which is the ratio between the CRPS of the analysis and the CRPS of the free run (i.e. the unassimilated state):

$$CRPSS = 1 - \frac{CRPS_{DA}}{CRPS_{free}}, \tag{26}$$

where $CRPS_{DA}$ (resp. $CRPS_{free}$) is the $CRPS$ score of the analysis (resp. the free run). When positive, the closer to one the CRPSS is, the more the DA process improves the estimation. When negative, the assimilation process degrades the estimation compared with the free run.

As pointed out in Bocquet and Sakov (2014), it is crucial to remind that the performances of a DA scheme depend on the metrics chosen to assess its quality. We chose the CRPS because it rigorously generalizes the notion of Mean Absolute Error (MAE) to stochastic predictions. In addition, the decomposition of the average CRPS can also provide information on the reliability and the resolution of the posterior ensemble (Hersbach, 2000).





## 3 Results

### 3.1 Comparison of methods

#### 3.1.1 Performances on moisture variable correction

The three DA methods are first tested on the nominal setup presented in Section 2.4. Figure 5 compares the CRPS time series of moisture variables assimilated by the three DA methods with the free run simulation on plot 10. The evolution of the CRPS for both the free run and the assimilated state is highly correlated to the precipitation time series. Rainfall events and following recession periods lead to peaks in the CRPS chronicle showing an increased level of uncertainty in the system in wet conditions. During the three first assimilation cycles (up to 576h), the iEnKS and the EnKF decrease the error in a limited

extend compared with the free run. More precisely, in this first part of simulation, corrections from their analysis steps become of nearly no effect after each rainfall event that follows an analysis. Until 576h, the ES-MDA is then the only method that allows for a clear reduction of the CRPS, both during rainfall events and dry periods. The second part of the simulation is characterised by longer and more intense rainfall events. In this time period all methods allow for an effective decrease of the model error. Broadly speaking, the ES-MDA leads to the lower CRPS values and the smaller discrepancies between dry and

wet periods showing comparable performances for both hydrological regimes.

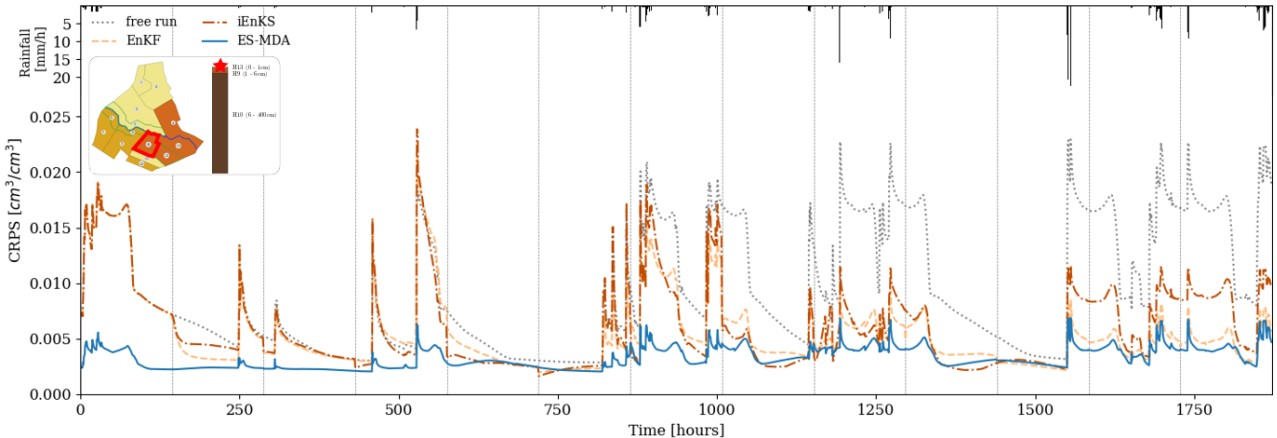

**Figure 5.** Comparison of CRPS time series for the free run and the different DA methods on plot 10 surface compartment. The position of plot 10 is depicted by the red line on the top-right map while the position of the surface cell in the associated soil column is depicted by a red star. The rainfall chronicle is shown in the top histogram while vertical grey lines correspond to times when surface moisture observations are available.

Unlike surface moisture estimation, the three methods exhibit much more limited performances in the subsurface as shown in Figure 6. At 0.2m depth, the soil only reacts to stronger rain events from 1,200h leading once again to peaks in the CRPS series within this time period. Again, the ES-MDA performs best to decrease the error although the gain from the free run is much more limited than for surface estimation. The iEnKS generally degrades the moisture estimation, especially between





1,000h and 1,250h. The iEnKS is the only method that is characterised by a moving DA windows. At each analysis, the current
state is corrected using the next $L$ observations ($L = 5$ in this case). The change in the model dynamics between the analysis
time step and the following observation time steps observed in the intermediary cell after 1,200h (see Figure 4, middle) may
explain such poor correction of the state by the iEnKS.

In the deeper cell, the CRPS remains constant between each analysis step (see Figure 4, bottom). The EnKF and the ES-MDA

also achieve a limited decrease in CRPS compared with the free run whereas the iEnKS once again degrades the estimation.
Note that from 1,152h, the iEnKS no longer performs analysis because there are not enough observations available in the DAW.
The system is no longer corrected from this moment onwards.

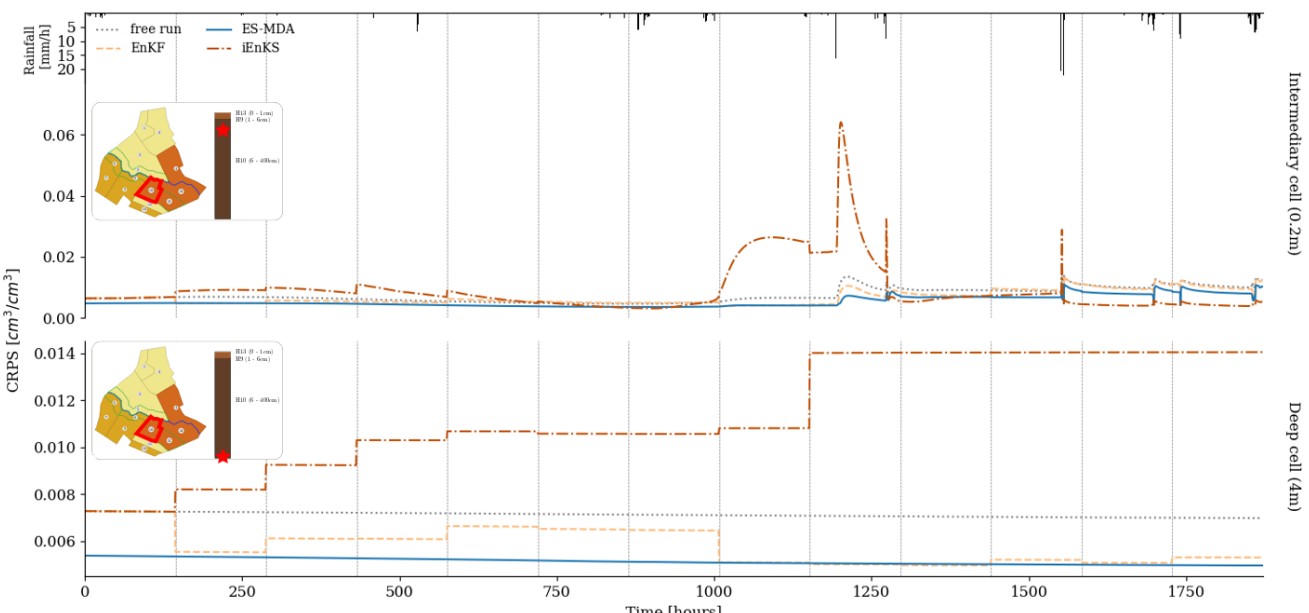

**Figure 6.** Comparison of CRPS time series for the free run and the different DA methods in intermediary cell (0.2m depth, top figure) and in
deeper cell (4m depth, bottom figure) for plot 10. The position of plot 10 is depicted by the red line on the top-right map while the position
of the cell in the associated soil column is depicted by a red star. The rainfall chronicle is shown in the top histogram while vertical grey lines
correspond to time steps when surface moisture observations are available.

    Average CRPSSs over the whole temporal period and the whole catchment at the scrutinized depths are computed for the
three methods in order to quantitatively identify the most appropriate DA method for the PESHMELBA model. All methods

significantly improve moisture estimation at surface with CRPSSs superior to 0.38 in average. The ES-MDA almost always
outperforms the EnKF and the iEnKS at surface but also, more slightly, at subsurface (Figure 7). Conversely, the EnKF signifi-
cantly degrades the state estimate in the mid cells for most of the UHs (negative CRPSS value). In the deeper cell, the ES-MDA
also outperforms the EnKF in average (0.15 vs 0.06) while the iEnKS leads to significantly increased error compared to the
free run. However, when looking locally, better CRPSSs are performed by ES-MDA on 9 UHs over the 14 ones, which makes



it impossible to generalize. Indeed, the contribution of assimilation is much more limited in the intermediary compartment which is not observed, and not significant in the deep compartment.

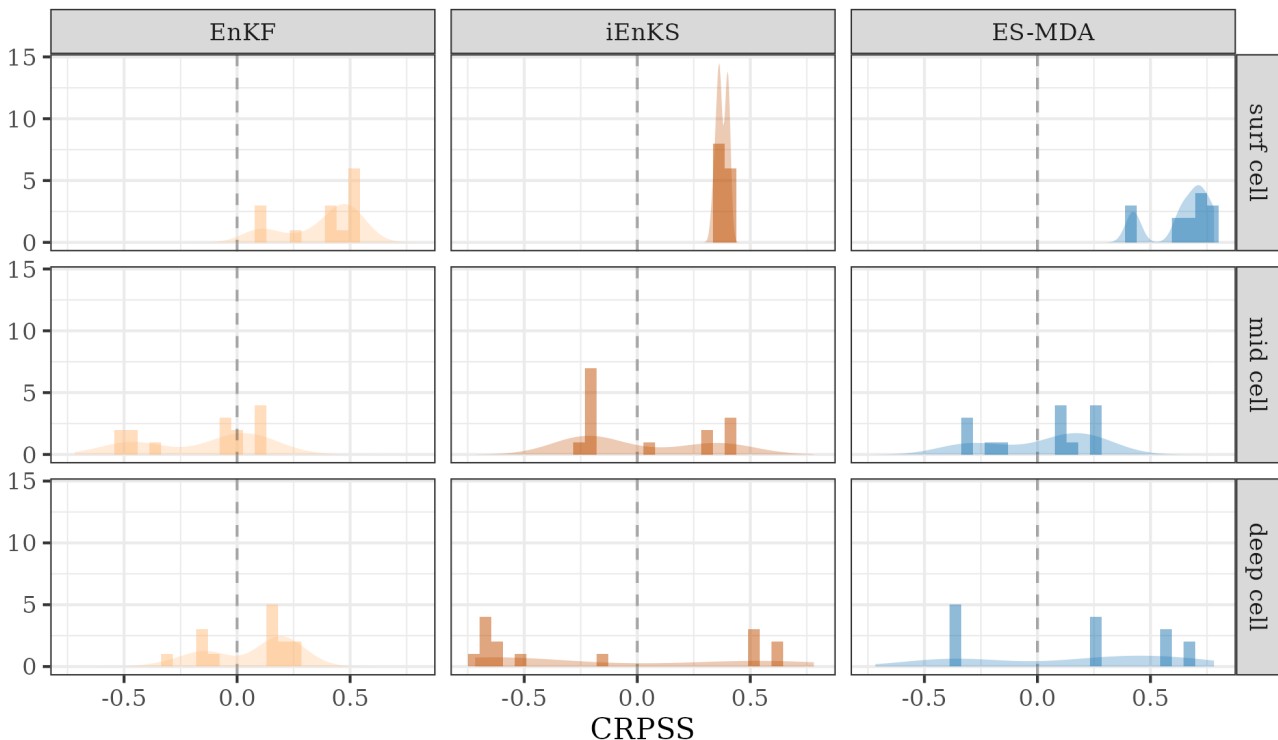

**Figure 7.** Distribution over the 14 HUs of CRPSS averaged over time. The vertical line at CRPSS = 0 highlights a decrease in error compared with the free run (positive values) or an increase in error compared to the free run (negative values).

### 3.1.2 Performances on parameter estimation

Performances on parameter estimation are also assessed and Figure 8 shows the posterior pdfs of estimated water contents at saturation $\theta_s$ while Table 2 gathers associated CRPSS values. Estimation of $\theta_s$ for surface horizons is significantly improved by

the three methods as CRPSS values exceed 0.58 in all cases. Still based on CRPSS values, the EnKF and the ES-MDA perform the best. Results are less clear-cut for estimation of subsurface parameters. Indeed, CRPSS values are far lower than for surface and the DA process even degrades $\theta_s$ estimation for horizon 4, 7 and 6 except for the iEnKS. For subsurface parameters, the iEnKS tends to perform the best although it once again achieves limited performances compared with surface parameter estimation. While the EnKF and the ES-MDA rely on the correlation matrix to estimate non observed parameters, the iEnKS

builds a cost function that explicitly relates to the input parameters. The full correlation matrix (not shown here) shows that there is little or no correlation between the moisture in the upper 5 cm and the parameters in the subsurface which may explain why the iEnKS performs better in such compartment. Finally, for both surface and subsurface parameters, the posterior pdf





and the associated CRPSS values got from the EnKF and the ES-MDA are quite close. It shows that assimilating observations one by one or all at once, using an ensemble Kalman filter based method, leads to comparable performances for parameter
estimation whereas significant differences are noticed for variable correction.

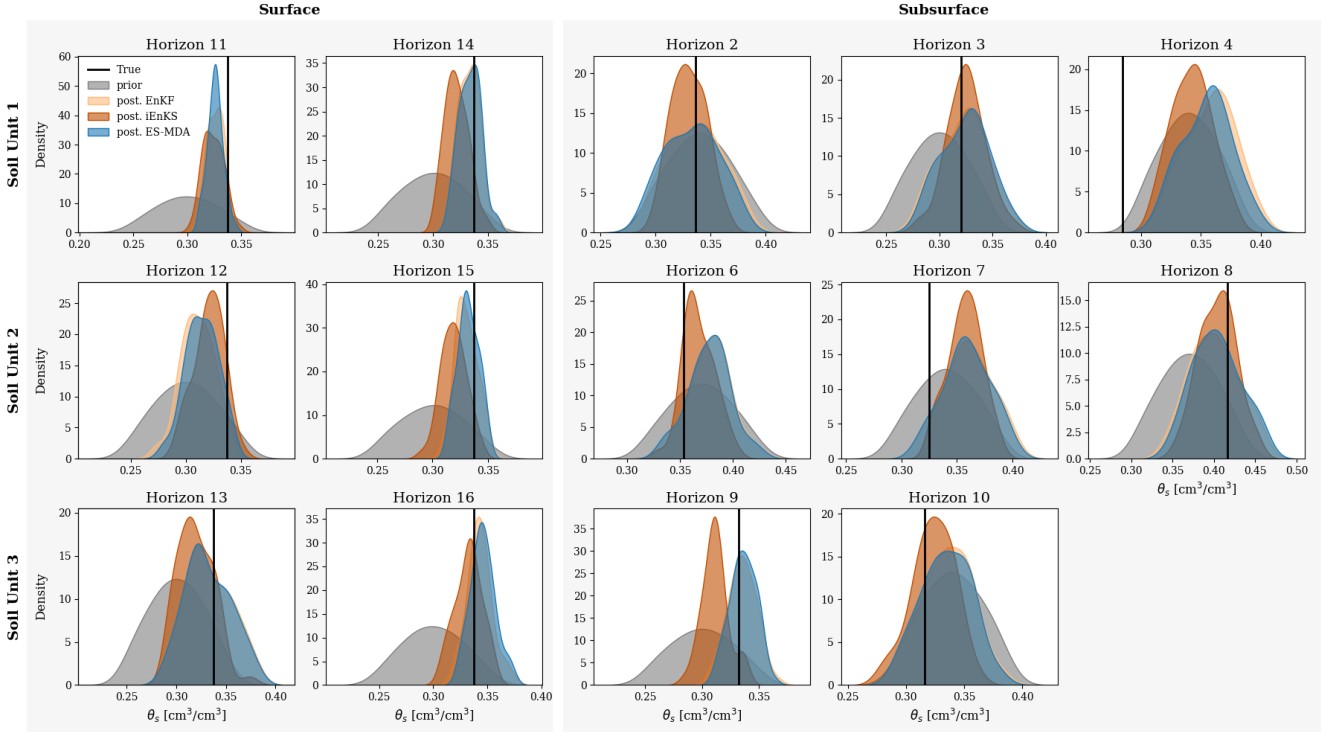

**Figure 8.** Empirical pdfs for estimated surface parameters for SU1 (top line), SU2 (middle line) and SU3 (bottom line). The first column gathers pdfs of surface $\theta_s$ on vineyard plots while the second column refers to surface parameters on VFSs. Remaining columns gather parameters for subsurface horizons. The true values of parameters are indicated by a vertical black line on each plot.

### 3.1.3  Computational cost

Computational costs of the three methods are rather contrasted. The EnKF is the fastest method (277 hCPU) followed by the ES-MDA (558 hCPU) and the iEnKS (2143 hCPU). Compared with the EnKF, both the ES-MDA and the iEnKS perform incremental corrections leading to such higher computational costs. Furthermore, in the case of the iEnKS, using an assimilation
window of size $L$ implies that the model must be integrated up to $M \times L \times jmax$ times to perform each analysis (see Table 6.1). As PESHMELBA integration is the most costly step in the assimilation process, the iEnKS computational cost ends up much more significant than for the other methods.





| | Horizon | EnKF | ES-MDA | iEnKS |
|---|---|---|---|---|
| | 11 | **0.68** | 0.65 | 0.62 |
| | 12 | 0.18 | 0.36 | **0.58** |
| Surface | 13 | **0.70** | **0.70** | 0.52 |
| | 14 | 0.87 | **0.88** | 0.54 |
| | 15 | 0.80 | **0.85** | 0.47 |
| | 16 | **0.86** | 0.80 | 0.85 |
| | 2 | 0.06 | 0.07 | **0.26** |
| | 3 | 0.51 | 0.48 | **0.64** |
| | 4 | -0.47 | -0.39 | **-0.14** |
| | 6 | -0.52 | -0.48 | **0.28** |
| Subsurface | 7 | -1.44 | **-1.35** | -1.42 |
| | 8 | 0.62 | 0.66 | **0.74** |
| | 9 | **0.84** | 0.83 | 0.23 |
| | 10 | 0.16 | 0.20 | **0.59** |

**Table 2.** CRPSS values associated to parameter estimation for the three DA methods. Positive values indicate a decrease in error compared with the free run while negative values indicate an increase in error compared with the free run. For each line, the bold value refers to the best estimate.

## 3.2 Sensitivity of the methods to DA setup

In this section, the sensitivity of the different methods to observation error prescription, assimilation frequency and ensem-

ble size are investigated. As the tested DA methods have shown limited performances to retrieve subsurface moisture from satellite surface moisture images, the analysis focuses on surface moisture estimation, following several scenarios (Table 1). Furthermore, as mentioned in the previous section, the high computational cost of the iEnKS is prohibitive to perform intensive exploration of the method in this case study. Sensitivity analysis is then only performed for the two methods with the highest potential in this application, EnKF and ES-MDA.

### 3.2.1 Observation error

The sensitivity to the observation error magnitude is tested on a setup with frequency of assimilation of 6 days (144 h) and ensemble size of 50 on a 78-day long simulation and evaluated on averaged CRPSS (Figure 9). As expected, for both methods the CRPSS is the highest for small observation errors and regularly decreases for larger observation errors. In the case of EnKF, no significant correction of the state can be obtained from error values superior to 0.1 $cm^3cm^{-3}$. In the case of ES-MDA,

moisture is noticeably corrected for error values up to 0.2 $cm^3cm^{-3}$.

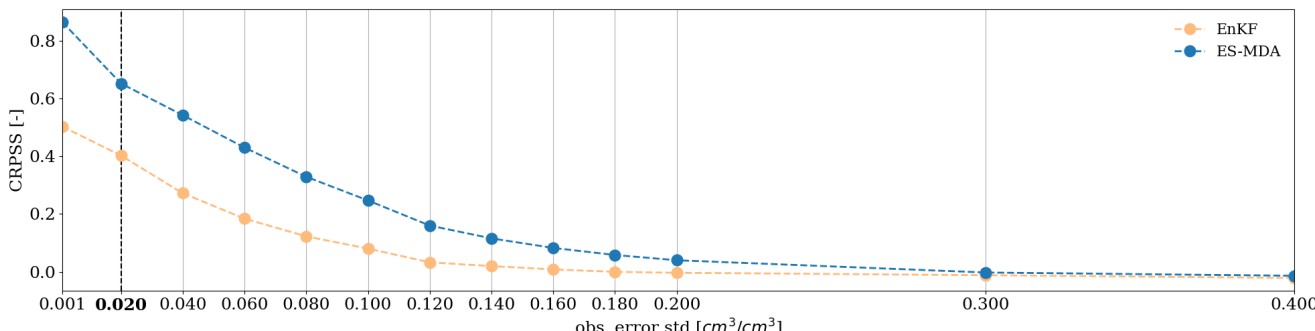

**Figure 9.** CRPSS sensitivity to standard deviation of observation error for the surface cell with EnKF and ES-MDA. The CRPSS is computed as an average over the whole time series and over the whole catchment. The bold label and vertical dashed line denote the nominal value of observation error used in this case study.

### 3.2.2 Frequency of observation

The averaged CRPSS in the surface compartment for the EnKF and the ES-MDA is also evaluated in function of the observation frequency (Figure 10). The observation error magnitude is set to 0.02 cm$^3$cm$^{-3}$ and the ensemble size is set to 50. In the case of EnKF, the CRPSS is relatively stable for observation frequencies (meaning analysis frequencies) up to 72h and decreases
regularly for lower frequencies. ES-MDA performances are rather stable for frequencies of observation inferior to 144h (6 days). Performances are slightly worst for frequencies of 192 h and 216 h, (even if still higher than the most performant EnKF setup) and drop significantly if observations are available only at each 240 h (10 days).

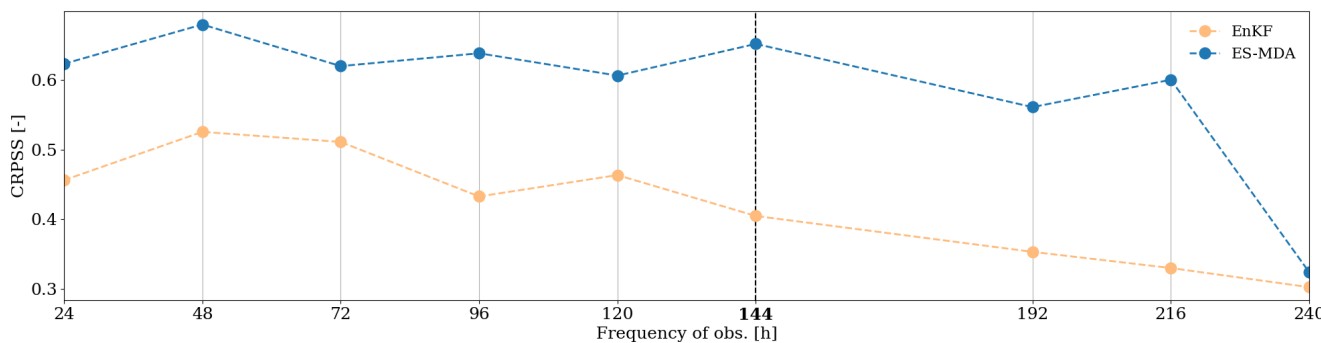

**Figure 10.** Comparison of the CRPSS sensitivity to observation frequency for the surface cell. The CRPSS is computed as an average over the whole time series and over the whole catchment. The bold label and vertical dashed line denote the nominal frequency of observation. This value also corresponds to the realistic frequency of observation for surface moisture images used in this case study.

### 3.2.3 Ensemble size

While the observation error magnitude and the observation frequency are intrinsic properties of the observation set in practice,
the ensemble size is a parameter of the DA setup that can be tuned by the user. Its choice also critically impacts the numerical





cost of the data assimilation and should be chosen to reach a satisfying trade-off between limited computational cost and suffi-
cient accuracy of the analysis. In this experiment, the sensitivity on the ensemble size is tested on a scenario with observation
error of 0.02 cm$^3$cm$^{-3}$ and with frequency of observations of 144h (Figure 11).

For both methods, the lower CRPSS value is reached for an ensemble size of 20. The EnKF performances stabilize for
ensemble size superior to 50 members while 100 members are necessary for the ES-MDA partly due to its higher state vector
size that includes temporal dimension.

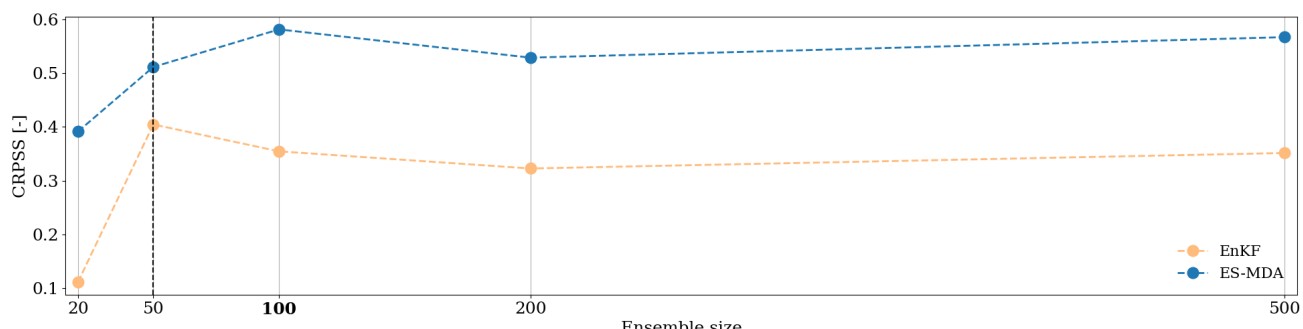

**Figure 11.** CRPSS sensitivity to the ensemble size for the surface cell. The CRPSS is computed as an average over the whole time series and
over the whole catchment. The bold label and vertical dashed line denote the nominal value of ensemble size used in this case study.

## 4  Discussion

### 4.1  On the choice of the DA method

Results from the above section indicate that in this case study, assimilating observations one at a time is sufficient for esti-
mating input parameters. However, integrating several observations simultaneously is more efficient for correcting moisture
trajectories, particularly when using the Ensemble Smoother with Multiple Data Assimilation. The ES-MDA better constrains
the system as it incorporates information from all observations and the system dynamics, making it especially effective in cop-
ing with nonlinearity, as noted by Emerick and Reynolds (2013a). The ES-MDA ensures that corrections can propagate from
observed to unobserved times, particularly from rainy periods to inter-event periods, given sufficient temporal correlations. In
contrast, during the initial dry hydrological regime, saturated water contents ($\theta_s$) are unobservable, limiting the performances
of the EnKF and the iEnKS, which correct the augmented state vector at a specific time. The global and iterative correction of
the ES-MDA allows the impact of accurate observations during wet periods to extend to unobservable dry periods. In addition,
although the iEnKS demonstrates potential for parameter estimation, it is clear that the additional computational cost for mois-
ture correction is not worth it. It is likely because PESHMELBA modular approach results in the linear coupling of vertical
1D Richards equation through lateral Darcy's law, rather than employing the nonlinear full 3D Richards equation. Such linear
coupling reduces the potential benefits on highly nonlinear systems of using iEnKS.





From the above, the ES-MDA is identifed as the best compromise for assimilating moisture images for joint estimation of moisture profiles and $\theta_s$ parameters in the PESHMELBA model. The ES-MDA performs the best for moisture correction and achieves performances comparable to the EnKF for parameter estimation. This method is also straightforward to implement and does not require frequent interruptions of the calculation code to perform the analysis. Additionally, although the ES-MDA is an iterative method, its computational cost remains reasonable, as only a few iterations are necessary to optimize its performances.

### 4.2 On the real-case DA implementation

The exploration of DA implementation scenarios (section 3.2) allows suggesting some practical guidelines to best adapt the setup to the PESHMELBA model. First, the quality of assimilated data plays a crucial role: results provide the user of moisture data obtained from Sentinel-1/Sentinel-2 synergistic inversion with guidelines on the required quality of future observations on vineyard cover. For example, one could keep in mind that an error on observations inferior to 0.05cm$^3$cm$^{-3}$ with the EnKF (resp. 0.1cm$^3$cm$^{-3}$ with the ES-MDA) will be required to reach significant improvements (superior to 20% compared with the free run) of surface moisture estimation. Regarding the frequency of data, it is set to 144h in average for surface moisture images computed from Sentinel-1 and Sentinel-2 satellite. In this application, this frequency is nearly optimal, other factors being equal, to maximize the performances of the ES-MDA for DA window lengths around 3 months, as all observations are integrated at the same time. In the case of the EnKF, a frequency of 72h would be necessary to optimize its performances.

Finally, the ensemble size of 100 members is advised for both methods to ensure a stable performance, as often chosen in studies that describe their DA experiment setup in hydrology (Camporese et al., 2009; Nie et al., 2011; Lei et al., 2020).

### 4.3 On the limitations of the methods

Even if the different DA methods succeed in estimating surface variables and parameters, the gain is limited to retrieve subsurface variables and parameters. Moisture observations only cover the top 5cm and none of the tested method properly retrieves deeper values. In theory, both the iEnKS and the ES-MDA that are characterised by longer DAW lengths may have been able to capture the subsurface dynamics if it had remained correlated to the surface one. However, the correlation matrices extracted at some time steps such as shown in Figure 12, and for the whole time series (not shown here) show that there are nearly no correlations between the surface and the subsurface even considering potential time lags. In this case study, the composition of each soil type with distinct horizons, as well as the vertical discretization, much more refined at surface than in the subsurface, may explain the lack of correlation. To be noted that such a lack of correlation has also been observed in Bonan et al. 2020 (in arid contexts only).

We conclude that assimilating observations of top soil moisture cannot properly correct subsurface variables or parameters. However, from the correlation matrix (Figure 12), we see that the moisture in a given soil horizon is highly correlated to moisture in this same soil horizon in other plots that also contain this specific soil horizon. It means that observations of a full soil column in some points in the catchment may be sufficient to correct all plots with the same soil type. This option must be further investigated, for example by assimilating measurements from electronic magnetic interference that can provide punctual





pseudo observations of soil moisture vertical profiles. In this case, we would need to introduce a new observation error model
since the moisture profiles would be deduced from resistivity measurements.

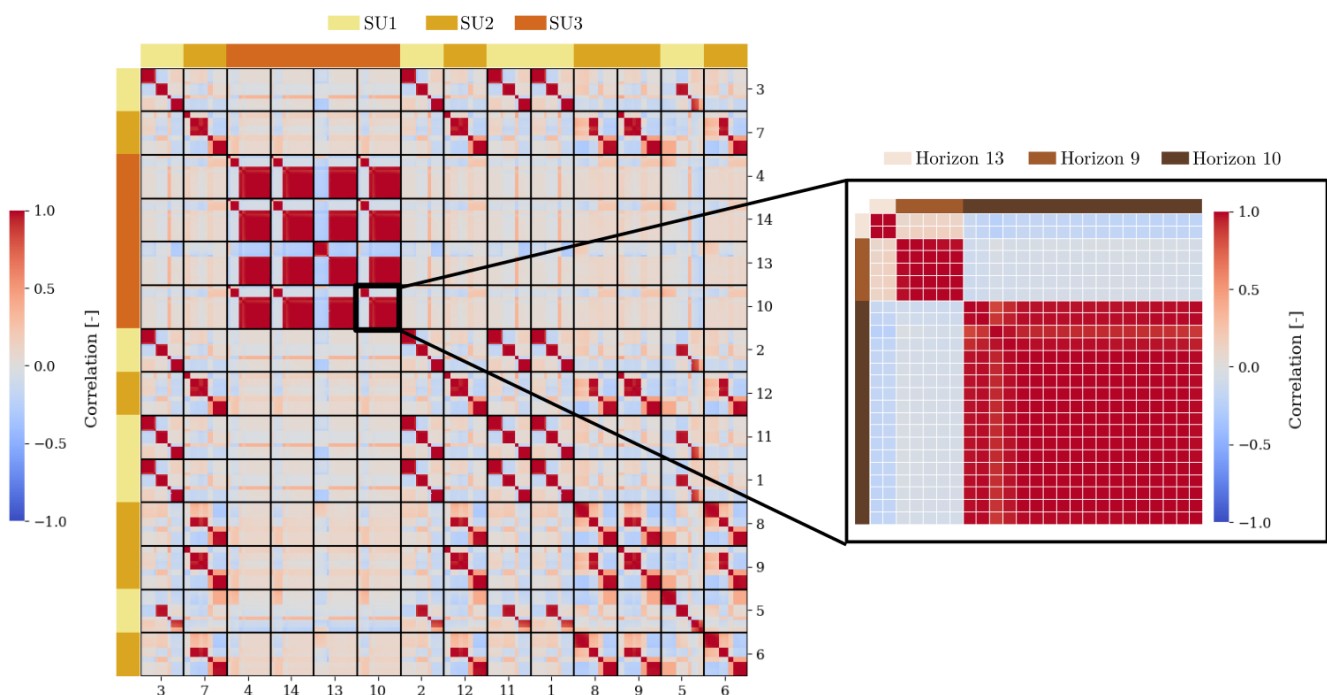

**Figure 12.** Left: Correlation matrix of the ensemble after 144h of simulation in the free run. Vertical and horizontal lines delimit the numerical
cells from a same plot. The top and left colors indicate the soil type for each plot while the bottom and right indices are the plot indices.
Right: Portion of the correlation matrix that relates to plot 10. Each square outlined in white refers to a numerical cell in the vertical profile
of plot 10. The top and left colors refer to the soil horizon for each numerical cell.

## 5 Conclusions

In this study, we have set a rigorous data assimilation framework for a modular model that is specifically built for water
quality management. The conducted experiments aimed at retrieving vertical moisture profiles but also at estimating some
input parameters, based on synthetic surface moisture satellite images. To do so, we have implemented several stochastic
ensemble methods, namely the Ensemble Kalman Filter, the Ensemble Smoother with Multiple Data Assimilation and the
iterative Ensemble Kalman Smoother. These three methods are representative of the spectrum of available ensemble methods
(ensemble filters, hybrid ensemble/variational smoothers and long window ensemble smoothers). They solve the DA problem
differently and in particular they use the available observations in different ways. We have then assessed and compared their
performances for joint estimation so as to choose the most appropriate method for the PESHMELBA model.
Results of comparison showed that all methods performed well to retrieve moisture and parameters at surface but the ES-



MDA significantly outperformed the EnKF and the iEnKS. The ES-MDA is indeed the only method that can fully integrate the system dynamics in its analysis step. Results for subsurface, however, showed that all methods failed in retrieving the rest of the vertical moisture profiles and associated sauturated water content parameters. The analysis of the correlation matrices showed

that the surface and the subsurface compartments are poorly correlated meaning that one should not expect surface moisture data alone to improve moisture estimate in deeper compartments. Furthermore, results of sensitivity analysis on observation error, observation frequency and ensemble size provided practical guidelines to implement an adapted DA setup in future operational applications.

As the proposed setup has been shown to be limited for retrieving subsurface moisture, further research could extend this study

to multi-source data assimilation. For instance, the joint assimilation of surface moisture images and punctual observations of vertical moisture profiles got from electronic magnetic interference should be further investigated to improve moisture estimation in deep soil compartments. In addition, when setting any stochastic DA experiment, the ensemble must be generated so as to represent the model error as accurately as possible. In this study, only the parameter values were considered as an uncertainty source which is certainly too simplistic. Future experiments should then integrate uncertainty on initial conditions

and climate forcings.

Systematically taking into account and reducing the uncertainties in risk assessment models is a major issue especially when these models have an operational impact. This study paves the way for further applications in risk assessment models, even based on a complex structure, to assess both water and pollutant transfers in agricultural landscapes.

*Code availability.* The PESHMELBA model is an open-source model coded in Python (Version 2.7.17) and Fortran 90 and embedded in the

OpenPALM coupler (Version 4.3.0). The code for the OpenPALM coupler is available from www.cerfacs.fr/globc/PALM_WEB/user.html# download after registration. The exact version of PESHMELBA so as Python (Version 3.7) and bash scripts used for data assimilation are provided in the following Zenodo repository: https://zenodo.org/record/6782073#.Yr1DmDXP2Uk.

## A  Input parameters description, nominal values and pdfs

| Input factor [units] | Description | Nominal Value | Pdf |
|---|---|---|---|
| Soil parameters for Horizon 11, 12 and 13 | | | |
| $thetas$ [m³m⁻³] | Saturated water content | 0.34 | N(0.34, 0.03) |
| $thetar$ [m³m⁻³] | Residual water content | 0.04 | TN(0.04, 9.3e-3, 0, 1) |
| $Ks$ [ms⁻¹] | Saturated hydraulic conductivity | 3.93e-5 | LN(-10.16,-2.03) |
| $hg$ [m] | Air-entry pressure in VG retention curve | -9.69e-2 | N(-9.69e-2,9.69e-3) |
| $mn$ | Deduced parameter from VG $n$ parameter | 0.27 | N(0.27, 2.7e-2) |





| | | | |
|---|---|---|---|
| $Ko$ [ms$^{-1}$] | Matching point at saturation in modified MVG conductivity curve | 2.86e-7 | LN(-15.09,-3.02) |
| $L$ | Empirical pore-connectivity parameter | -8.43 | U(-10.11, -6.74) |
| $bd$ [gcm$^{-3}$] | Bulk density | 1.34 | U(1.08, 1.61) |
| Soil parameters for Horizon 14, 15 and 16 | | | |
| $thetas$ [m$^3$m$^{-3}$] | Saturated water content | 0.34 | N(0.34, 0.03) |
| $thetar$ [m$^3$m$^{-3}$] | Residual water content | 0.04 | TN(0.04, 9.3e-3, 0, 1) |
| $Ks$ [ms$^{-1}$] | Saturated hydraulic conductivity | 4.31e-5 | LN(-10.11, -2.02) |
| $hg$ [m] | Air-entry pressure in VG retention curve | -9.69e-2 | N(-9.69e-2, 9.69e-3) |
| $mn$ | Deduced parameter from VG $n$ parameter | 0.27 | N(0.27, 2.7e-2) |
| $Ko$ [ms$^{-1}$] | Matching point at saturation in modified MVG conductivity curve | 2.86e-7 | LN(-15.09, -3.02) |
| $L$ | Empirical pore-connectivity parameter | -8.43 | U(-10.11, -6.74) |
| $bd$ [gcm$^{-3}$] | Bulk density | 1.34 | U(1.08, 1.61) |
| Soil parameters for Horizon 2 | | | |
| $thetas$ [m$^3$m$^{-3}$] | Saturated water content | 0.34 | N(0.34, 0.03) |
| $thetar$ [m$^3$m$^{-3}$] | Residual water content | 0.05 | TN(0.05, 0.01, 0, 1) |
| $Ks$ [ms$^{-1}$] | Saturated hydraulic conductivity | 8.64e-5 | LN(-9.38, -1.88) |
| $hg$ [m] | Air-entry pressure in VG retention curve | -3.29e-2 | N(-3.29e-2, 3.29e-3) |
| $mn$ | Deduced parameter from VG $n$ parameter | 0.2 | N(0.2, 2e-2) |
| $Ko$ [ms$^{-1}$] | Matching point at saturation in modified MVG conductivity curve | 2.28e-7 | LN(-15.31,-3.06) |
| $L$ | Empirical pore-connectivity parameter | -6.52 | U(-7.82, -5.21) |
| $bd$ [gcm$^{-3}$] | Bulk density | 1.47 | U(1.18, 1.77) |
| Soil parameters for Horizon 3 | | | |
| $thetas$ [m$^3$m$^{-3}$] | Saturated water content | 0.32 | N(0.32, 0.03) |
| $thetar$ [m$^3$m$^{-3}$] | Residual water content | 0.08 | TN(0.08, 0.02, 0, 1) |
| $Ks$ [ms$^{-1}$] | Saturated hydraulic conductivity | 5.39e-5 | LN(-9.85, -1.97) |
| $hg$ [m] | Air-entry pressure in VG retention curve | -2.09e-2 | N(-0.02, 2.1e-2) |



| | | | |
|---|---|---|---|
| $mn$ | Deduced parameter from VG $n$ parameter | 0.20 | N(0.2, 2e-2) |
| $Ko$ [ms⁻¹] | Matching point at saturation in modified MVG conductivity curve | 7.47e-7 | LN(-14.13,-2.83) |
| $L$ | Empirical pore-connectivity parameter | -4.24 | U(-5.08, -3.39) |
| $bd$ [gcm⁻³] | Bulk density | 1.57 | U(1.25, 1.88) |
| **Soil parameters for Horizon 4** | | | |
| $thetas$ [m³m⁻³] | Saturated water content | 0.28 | N(0.28, 0.03) |
| $thetar$ [m³m⁻³] | Residual water content | 0.07 | TN( 0.07, 0.02, 0, 1) |
| $Ks$ [ms⁻¹] | Saturated hydraulic conductivity | 3.11e-5 | LN(-10.40,-2.08) |
| $hg$ [m] | Air-entry pressure in VG retention curve | -5.99e-2 | N(-6e-2, 6e-3) |
| $mn$ | Deduced parameter from VG $n$ parameter | 0.23 | N(0.23, 0.02) |
| $Ko$ [ms⁻¹] | Matching point at saturation in modified MVG conductivity curve | 1.47e-6 | LN(-13.45, -2.69) |
| $L$ | Empirical pore-connectivity parameter | -0.14 | U(-0.17, -0.11) |
| $bd$ [gcm⁻³] | Bulk density | 1.53 | U(1.22, 1.84) |
| **Soil parameters for Horizon 6** | | | |
| $thetas$ [m³m⁻³] | Saturated water content | 0.35 | N(0.35, 0.04) |
| $thetar$ [m³m⁻³] | Residual water content | 0.01 | TN(0.01, 9.3e-3, 0, 1) |
| $Ks$ [ms⁻¹] | Saturated hydraulic conductivity | 2.16e-5 | LN(-10.77, -2.15) |
| $hg$ [m] | Air-entry pressure in VG retention curve | -6.60e-2 | N(0.07, 6.60e-3) |
| $mn$ | Deduced parameter from VG $n$ parameter | 0.13 | N(0.13, 1.3e-2) |
| $Ko$ [ms⁻¹] | Matching point at saturation in modified MVG conductivity curve | 3.19e-7 | LN(-14.97, -3.00) |
| $L$ | Empirical pore-connectivity parameter | 9.66 | U(7.72, 19.31) |
| $bd$ [gcm⁻³] | Bulk density | 1.59 | U(1.27, 1.91) |
| **Soil parameters for Horizon 7** | | | |
| $thetas$ [m³m⁻³] | Saturated water content | 0.32 | N(0.32, 0.03) |
| $thetar$ [m³m⁻³] | Residual water content | 0.01 | TN(0, 9.31e-3, 0, 1) |
| $Ks$ [ms⁻¹] | Saturated hydraulic conductivity | 9.60e-6 | LN(-11.57, -2.31) |



| | | | |
|---|---|---|---|
| $hg$ [m] | Air-entry pressure in VG retention curve | -7.18e-2 | N(-0.07, 7.18e-3) |
| $mn$ | Deduced parameter from VG $n$ parameter | 0.08 | N(0.08, 7.5e-3) |
| $Ko$ [ms$^{-1}$] | Matching point at saturation in modified MVG conductivity curve | 1.67e-7 | LN(-15.63, -3.13) |
| $L$ | Empirical pore-connectivity parameter | -10 | U(-12, -8) |
| $bd$ [gcm$^{-3}$] | Bulk density | 1.66 | U(1.33, 1.99) |
| **Soil parameters for Horizon 8** | | | |
| $thetas$ [m$^3$m$^{-3}$] | Saturated water content | 0.42 | N(0.42, 0.04) |
| $thetar$ [m$^3$m$^{-3}$] | Residual water content | 0.01 | TN(0, 9.3e-3, 0, 1) |
| $Ks$ [ms$^{-1}$] | Saturated hydraulic conductivity | 3.98e-6 | LN(-12.45, -2.49) |
| $hg$ [m] | Air-entry pressure in VG retention curve | -0.30 | N(-0.30, 3.02e-2) |
| $mn$ | Deduced parameter from VG $n$ parameter | 0.08 | N(0.08, 8e-3) |
| $Ko$ [ms$^{-1}$] | Matching point at saturation in modified MVG conductivity curve | 9.72e-8 | LN(-16.17, -3.23) |
| $L$ | Empirical pore-connectivity parameter | 10 | U(8, 12) |
| $bd$ [gcm$^{-3}$] | Bulk density | 1.54 | U(1.23, 1.85) |
| **Soil parameters for Horizon 9** | | | |
| $thetas$ [m$^3$m$^{-3}$] | Saturated water content | 0.33 | N(0.33, 0.03) |
| $thetar$ [m$^3$m$^{-3}$] | Residual water content | 0.08 | TN(0.08, 1.92e-2, 0, 1) |
| $Ks$ [ms$^{-1}$] | Saturated hydraulic conductivity | 3.05e-5 | LN(-10.41, -2.08) |
| $hg$ [m] | Air-entry pressure in VG retention curve | -6.71e-2 | N(-6.71e-2, 6.71e-3) |
| $mn$ | Deduced parameter from VG $n$ parameter | 0.26 | N(0.26, 0.03) |
| $Ko$ [ms$^{-1}$] | Matching point at saturation in modified MVG conductivity curve | 3.36e-7 | LN(-14.93, -2.99) |
| $L$ | Empirical pore-connectivity parameter | 0.42 | U(0.34, 0.84) |
| $bd$ [gcm$^{-3}$] | | Bulk1.46 density | U(1.17, 1.75) |
| **Soil parameters for Horizon 10** | | | |
| $thetas$ [m$^3$m$^{-3}$] | Saturated water content | 0.32 | N(0.32, 0.03) |



| Parameter | Description | Value | Distribution |
|---|---|---|---|
| $thetar$ [m$^3$m$^{-3}$] | Residual water content | 0.06 | TN(0.06, 0.02, 0, 1) |
| $Ks$ [ms$^{-1}$] | Saturated hydraulic conductivity | 2.38e-5 | LN(-10.67, -2.13) |
| $hg$ [m] | Air-entry pressure in VG retention curve | -3.56e-2 | N(-3.56e-2, 3.56e-3) |
| $mn$ | Deduced parameter from VG $n$ parameter | 0.18 | N(0.18, 0.02) |
| $Ko$ [ms$^{-1}$] | Matching point at saturation in modified MVG conductivity curve | 3e-7 | LN(-15.04, -3.01) |
| $L$ | Empirical pore-connectivity parameter | 1.05 | U(0.84, 2.09) |
| $bd$ [gcm$^{-3}$] | Bulk density | 1.62 | U(1.30, 1.95) |
| Vegetation parameters for vineyard | | | |
| $manning$ [sm$^{-1/3}$] | Manning's roughness | 3.3e-2 | T(2.5e-2, 3.3e-2, 4.1e-2) |
| $Zr$ [m] | Rooting depth | 2.62 | U(2.10,3.14) |
| $F10$ | Fraction of the root length density in the top 10% of the root zone | 0.37 | U(0.3, 4.44e-1) |
| $LAImin$ | Min LAI value | 0.01 | U(8e-3, 1.2e-2) |
| $LAImax$ | Max LAI value | 2.5 | U(2, 3) |
| $LAIharv$ | LAI value at harvest time | 0.01 | U(8e-3, 1.2e-2) |
| Vegetation parameters for VFZ | | | |
| $manning$ [sm$^{-1/3}$] | Manning's roughness | 0.2 | T(0.1, 0.2, 0.3) |
| $Zr$ [m] | Rooting depth | 0.9 | U(0.72, 1.08) |
| $F10$ | Fraction of the root length density in the top 10% of the root zone | 0.34 | U(0.27, 0.40) |
| $LAI$ | Constant LAI value | 5 | U(4, 6) |
| River parameters | | | |
| $hpond$ [m] | Ponding height in the river bed | 0.01 | U(8e-3, 1.2e-2) |
| $di$ [m] | Distance between the river bed and the limit of impervious saturated zone | 1.5 | U(1.2, 1.8) |
| $Ks$ [ms$^{-1}$] | Saturated conductivity of the river bed | 2.38e-5 | LN(-10.67, -2.13) |



| manning [sm$^{-1/3}$] | River bed Manning's roughness | 7.9e-2 | T(6.1e-2, 7.9e-2, 9.7e-2) |
|---|---|---|---|
| **Plot and VFS parameters** | | | |
| $hpond$ [m] | Ponding height on vineyard plot | 0.01 | U(8e-3, 1.2e-2) |
| $hpond$ [m] | Ponding height on VFZ | 0.05 | U(0.04, 0.06) |
| $adsorpthick$ [m] | Mixing layer thickness | 0.01 | U(5e-3, 1.5e-2) |

## B Evaluation of the CRPS

Assuming an ensemble of $M$ moisture values $Y_1,...,Y_M$ sorted from smallest to largest so that:

$$Y_i \leq Y_j, \text{ for } i < j, \tag{B1}$$

and $\hat{\theta}$ the deterministic reference value, the CRPS for the ensemble is evaluated as follows (Hersbach, 2000):

$$CRPS = \sum_{i=0}^{M} \alpha_i p_i^2 + \beta_i (1 - p_i)^2, \tag{B2}$$

where $p_i = \frac{1}{M}$ and the $\alpha_i$ and $\beta_i$ are defined as follows:

| $0 < i < M$ | $\alpha_i$ | $\beta_i$ |
|---|---|---|
| $\hat{\theta} > Y_i$ | $Y_{i+1} - Y_i$ | $0$ |
| $Y_{i+1} > \hat{\theta} > Y_i$ | $\hat{\theta} - Y_i$ | $Y_{i+1} - \hat{\theta}$ |
| $\hat{\theta} < Y_i$ | $0$ | $Y_{i+1} - Y_i$ |

$$\tag{B3}$$

The cases $i = 0$ and $i = M$ only participate to the CRPS when the reference value $\hat{\theta}$ is an outlier, meaning inferior to $Y_1$ or superior to $Y_M$. In this case, Table B3 must be modified with the following:

| Outlier | $\alpha_i$ | $\beta_i$ |
|---|---|---|
| $\hat{\theta} < Y_1$ | $0$ | $Y_1 - \hat{\theta}$ |
| $Y_M < \hat{\theta}$ | $\hat{\theta} - Y_M$ | $0$ |

$$\tag{B4}$$

*Author contributions.* All authors contributed to writing the text and to all stages of editing. Data assimilation experiments were conducted by Emilie Rouzies with extensive support from Claire Lauvernet and Arthur Vidard.





*Competing interests.* The authors declare that they have no conflict of interest.

*Acknowledgements.* The authors kindly acknowledge Jérémy Verrier for his support on HIICS cluster usage.





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
