# Peer review of "Comparison of ensemble assimilation methods in a decision support model for landscape management to mitigate pesticide transfer"

_Hydrology and Earth System Sciences, 2024_

## Author Comment (AC1)

**RC1**: 'Comment on hess-2024-219', Benjamin Mary, 03 Oct 2024 reply

**Comparison of different ensemble assimilation methods in a modular hydrological model dedicated to water quality management**

*Emilie Rouzies1, Claire Lauvernet1, and Arthur Vidard2*

I have greatly appreciated the opportunity to read the article by Rouzies et al. The paper presents a valuable synthetic study, effectively illustrating the strengths and weaknesses of the EnKF method, both sequentially and with smoothing over observations, which in this case mimic remote sensing of soil water content. The article is well-written, with clear explanations of the methodology and model used.

While I have some reservations about relying on Soil Moisture Remote Sensing products, this study clearly highlights the challenges associated with using such data to calibrate subsoil and plant parameters.

We thank very much the reviewer 1 for his positive comments on our study and took into account most of his comments, improving the paper.

Despite this, I believe the overall objectives of the study are achieved, particularly in terms of selecting the most suitable data assimilation (DA) scheme. However, I think a more detailed analysis of the DA results would strengthen the paper. For example, it would be beneficial to explore the following points:

- The choice of DA localization schemes, either through local domain DA or covariance localization, particularly given the variability in soil units (see, for instance, https://www.frontiersin.org/journals/applied-mathematics-and-statistics/articles/10.3389/fams.2019.00003/full).

Thank you for this reference and comment. Indeed, localization schemes could be implemented by using the covariance localization or local domain DA . About the covariance, this would be feasible with another implementation of the Kalman Filter, but not with ETKF since the covariance matrix is not built explicitly in this method. About the local domain DA, this would be very interesting and quite relevant, especially considering the structure of soils that are described (see Figure 12 that shows the spatial correlation by soil type).

In the discussion, l. 456, we added this sentence :

l 456. Figure 12 also highlights the absence of spatial correlations between soil units of different soils, advocating for using a scheme with local domain DA in ES-MDA (Asch et al., 2016) to alleviate the computational cost of this method that uses high dimension matrices.

 - The distinction between DA with only state updates versus DA that incorporates a training and validation phase (see Botto et al., 2018, www.hydrol-earth-syst-sci.net/22/4251/2018/).

In Botto et al. 2018, observations were assimilated only during the first 5 days of simulation, leaving the final 7 days as a validation period, during which the ensemble was left to evolve freely.

This approach in Botto et al. 2018 is interesting, however, this is not possible with ES-MDA because of the smoothing approach of the method. To compare the 3 methods, we made the choice to keep the same setup.

- Another important consideration is the number of parameters being updated. I understand that a previous sensitivity analysis was conducted on the same site. Would it have been advantageous to reduce the number of parameters updated based on the results of this prior analysis? (For reference: Global sensitivity analysis of the dynamics of a distributed hydrological model at the catchment scale).

Indeed, the model depends on a much higher number of parameters (128) than the ones that are updated in the DA scheme (14) . As noted in the paper (l. 285-290) : "Joint estimation is performed in order to estimate both vertical moisture profiles and relevant uncertain input parameters. The global sensitivity analysis of PESHMELBA in this case study [*performed with Polynomial Chaos Expansion to estimate Sobol indices on the dynamical outputs*] showed that parameters that most influence moisture profiles are mainly θs (water contents at saturation) for the different soil horizons (Radišić et al., 2023). The augmented state vector thus includes such parameters for both surface and deeper soil horizons and a bias is added to their pdfs when generating the initial ensemble." We added the number of updated parameters selected after this GSA in the paper (l.289).

Overall, I believe the manuscript holds strong potential for publication in HESS, pending major or minor revisions and further clarifications.

**SPECIFIC COMMENTS**

**--------------------------------**

The title could be more specific. I suggest rephrasing "modular hydrological model dedicated to water quality management," which is somewhat broad, to something more aligned with the core focus of the study, such as "Twin Experiments on a Virtual Catchment with Vegetative Filter Strips" or "Hydrology of Agricultural Catchments." This would better target the intended audience and improve the visibility of the paper in bibliographic searches.

Thank you for this advice, we changed "Comparison of different ensemble assimilation methods in a modular hydrological model dedicated to water quality management" to

"Comparison of ensemble assimilation methods in a decision support model for landscape management to mitigate pesticide transfer", we hope this suits your opinion. It also addresses the comment from RC2.

**Abstract**

L8: "some input parameters" specify them if possible => done

L9: "related input parameters" again specify (I suppose mainly VGP and plant traits?) => done

**Introduction**

L14-19: add references => done

L17: it is not clear what the authors means by "often simulate several physical processes" please rephrase

=> the sentence was modified to: Such hydrological models simulate several interacting physical processes in order to properly capture the complex reality of the field, such as surface and subsurface hydrological transfer, sediments, and pollutants among different units of the catchment.

L18: "They need large sets of input parameters" specify i.e. hydraulic conductivities, soil and vegetation physical properties, atmospheric boundary conditions, ..  => done

L21: rephrase " .. from observations distributed in time and space and PREDICTIONS from a numerical model". In DA the model is used for predicting.

=> changed with "simulations" : to us, the model is not always used for predictions (for example for a historical reanalysis)

L24: I would not say in geophysics but more in Earth Sciences (originally used for ocean modeling and weather forecasting)

=> what we meant was that Ensemble Kalman based methods are the most used among DA applications in hydrology. We changed with "Earth Sciences"

L24: Please rephrase "They consist of Monte Carlo algorithms and linear solutions of the estimation problem"

=> we change with "They use an approximation of the Kalman filter, where error statistics (typically mean and covariance matrix), are derived from an ensemble of members."

L25: I suggest rephrasing: an ensemble of realizations (instead of vectors) and adding i.e. by approximating the state by a state mean and covariance matrix. (if you finally want to keep the terminology "vectors" provide a reference to understand where it is from).

=> we change with "They use an approximation of the Kalman filter, where error statistics (typically mean and covariance matrix), are derived from an ensemble of members."

L27: as they come -replace by-> sequentially (?) => done

L28:  specify where the Gaussian assumption applies: on the assumption of Gaussian statistics (i.e. with forecast and measurement error distributions to be Gaussian). Reference: https://doi.org/10.1016/j.advwatres.2012.06.009 => done

L27-28: I suggest specifying that integrated hydrological models based on the Richards equation still represent a challenge, due to strong nonlinearities that may significantly affect the filter performance.

=> added : "In particular, assimilating in an integrated hydrological modeling based on the Richards equation still represents a challenge, due to strong nonlinearities that may significantly affect the filter performance."

L75: Please provide a reference that shows the following statement: "a filter, a hybrid variational/ensemble smoother that is efficient over short data assimilation windows and an ensemble smoother that is efficient over long data assimilation windows". Or maybe this is your assumption? by the way what timeline is consider long/short windows i.e days, years?? In any case explain. Thanks!

Long and short windows are relative to the model time steps, in this case a few days vs a few months. We added the reference of Asch, M., Bocquet, M., and Nodet, M.: Data assimilation: methods, algorithms, and applications, Fundamentals of Algorithms, SIAM, 2016.515.

**2.1 Model description**

L97 to 110: very interesting thanks for the detailed explanation of the model. Thanks!

**2.2 Data assimilation methods**

L120: replace/specify which parameters: "some input parameters"

Indeed, this is specified now, these are the van Genuchten soil water retention properties of all soil horizons, which pdfs are all described in Appendix A.

L135: can the author specify a reference for the inclusion of "an evolution law for the estimated parameters in addition to the state dynamical evolution".

This is implicit in most papers, but we suggest G. Evensen, "The ensemble Kalman filter for combined state and parameter estimation," in IEEE Control Systems Magazine, vol. 29, no. 3, pp. 83-104, June 2009, doi: 10.1109/MCS.2009.932223  (added in the text)

However, note that in this study, as soil characteristics are not expected to change over time at the scale of interest, we have chosen a persistence law to represent the (non-)evolution of the parameters.

**2.3 Case study**

- Size, superficie of the catchement?

This is a virtual catchment extracted from a real experimental site in Beaujolais region, France which is well described in Rouzies et al., 2023. As shown in the following figure, the subcatchment is quite small (~9 ha) and composed of 10 vineyard plots, four vegetative filter strips and five river reaches. We added the size in the text.

[Figure]

- Figure 4: are there important differences in surface/subsurface hydrology between the different Soil Units (SU)?

If so can you add another SU to fig 4.

If not could you mention it in the text?

We added the following sentence : "Three soil units (SU), mainly sandy and exhibiting fairly similar hydrological behaviors, compose the catchment in accordance with the soil composition of La Morcille catchment.."

Typically, when examining soil characteristics in the Appendix, the magnitudes are quite similar, particularly for the Van Genuchten parameters and Ksat. However, we choose to explicitly differentiate them to ensure that the Data Assimilation method remains applicable to other soil units that may be more distinct.

 In my perception, it is important to understand SU unit dynamics for the DA. If they are very different from each other then DA with localization (see comment below).

absolutely, and we answer to that below.

**2.4 DA setup**

 L273: I understand the idea of the TWIN experiment and using the True model to generate the observations. Something I'm not sure to understand is the spatial distribution of the observations. Are there several/one observations for each vineyard plot and VFS in the catchment? Are those gridded regularly? do you pick the mean for each zone?

In PESHMELBA, one originality is that it is not based on a classical meshing, but based on the landscape organization and on the concept of hydrological unit. So one plot is one unit/mesh and there is one observation per plot and per VFZ in the catchment.

Have you thought about Localization using the Local Analysis (LA) scheme for the different SU? The idea is to perform by spatially limiting the assimilation process within a certain distance from a grid point. read for instance: https://doi.org/10.1016/j.advwatres.2020.103813

 In any case, it would be interesting to analyze/discuss it in the text.

This comment was made above: Indeed, localization schemes could be implemented by using the covariance localization or local domain DA . About the covariance, this would be feasible with another implementation of the Kalman Filter, but not with ETKF since the covariance matrix is not built explicitly in this method. About the local domain DA, this would be very interesting and quite relevant, especially considering the structure of soils that are described (see Figure 12 that shows the spatial correlation by soil type).

In the discussion, l. 456, we added this sentence :

l 456. Figure 12 also highlights the absence of spatial correlations between soil units of different soils, advocating for using a scheme with local domain DA in ES-MDA (Asch et al., 2016) to alleviate the computational cost of this method that uses high dimension matrices.

L284: Have you considered the mutual correlation between the Van Genuchten parameter? (according to Carsel and Parrish (1988), who described their statistics and transformed them into normally distributed variables via the Johnson system (Johnson, 1970)? https://doi.org/10.1029/WR024i005p00755

Indeed, the standard approach for van Genuchten (VG) parameters typically assumes minimal interdependence, often disregarding potential relationships, such as the site-specific Kozeny-Carman (KC) equation. Note the use of "interdependence," which more accurately conveys the issue, as assuming a linear correlation (e.g., Pearson) is overly restrictive. It's important to

recall that these equations were initially proposed under the assumption of parameter independence, except in specific cases like Mualem's with m=1−1/n. Therefore, any empirical interdependence observed among parameters is not inherent to these empirical relationships.

However, for certain soils and conditions, measurements and statistical analyses of these parameters have generally revealed weak interdependencies among some of them. Managing weak interdependencies is challenging, even with advanced sampling methods like conditional probability schemes. Some of these interdependencies, well-known for a long time, reflect the physical properties of the soil. For instance, porosity and saturated hydraulic conductivity (Ks) are known to be related through a power function (the Kozeny-Carman equation). Nevertheless, this relationship is specific to certain soils and cannot be universally applied. I must thank Marnick Vanclooster and Rafael Muoz-Carpena with whom I discussed a lot of these issues. In Regalado and Muñoz-Carpena (2004), for example, they attempted to generalize this by converting the original KK equation into an stochastic form. In Lambert et al. 2025, we also explored this issue but from the source, in the way of sampling the VG parameters to perform a global sensitivity analysis of a model to design vegetative filter strips over France. In this study, we don't consider this dependence for all the reasons said above.

C. M. Regalado and R. Muñoz-Carpena, "Estimating the Saturated Hydraulic Conductivity in a Spatially Variable Soil With Different Permeameters: A Stochastic Kozeny–Carman Relation," Soil and Tillage Research 77, no. 2 (2004): 189–202, https://doi.org/10.1016/j.still.2003.12.008.

Lambert, G., Helbert, C. and Lauvernet, C. (2025), Quantization-Based Latin Hypercube Sampling for Dependent Inputs With an Application to Sensitivity Analysis of Environmental Models. Appl Stochastic Models Bus Ind. https://doi.org/10.1002/asmb.2899

**3.1.1 Performances on moisture variable correction**

Fig 5. I wonder if the 2nd part works better because the rain events are stronger or just because the parameters were already calibrated for a certain period and the model is thus already calibrated.

There are two ways of testing it:

- run without parameters update

- run with DA until time 1000h for instance and then let the system free

In any case, it would be interesting to analyze/discuss it in the text.

Indeed, this is a good remark. Note, however, that this figure shows the averaged behavior and only on UH 10, but as shown in Fig. 7, local behavior are a bit more complex to generalize. CRPSSs for some HU of the 14 ones can also be as good for the EnKF at surface.

ES-MDA is better from the beginning, on dry and wet periods, because of its integration of all observations at the same time. It gives enough temporal correlations to propagate from observed to unobserved times, so to rainy periods to inter-event periods. In the early stages of the dry hydrological period, saturated water contents cannot be observed, which constrains the effectiveness of the EnKF and iEnKS methods. These methods aim to adjust the augmented state vector at a particular time, but their performance is hindered by the lack of observable saturated water contents. We discuss it in the discussion parts 4.1 and 4.2.

**3.1.3 Computational cost**

L372: for how many ensembles that hCPU where calculated?

These are based on one ensemble of 50 members. For the iEnKS, the number of iterations to minimize the cost function was limited to 3 to make the computational cost reasonable compared to the other methods, with a moving assimilation window of 5 observations, and 3 iterations for ES-MDA. Considering these DA methods parametrization, the following table gives the number of simulations for each method, with M the number of ensemble members, C the number of assimilation cycles (forecast + analysis) in the simulation, J the number of iterations in the ES-MDA, L the assimilation window size for the iEnKS, and jmax the maximum number of iterations allowed for the iEnKS.

| | |
|---|---|
| EnKF | $MC$
 $50 \times 13$ |
| ES-MDA | $MJ$
 $50 \times 3$ |
| iEnKS | $M(C + Ljmax)$
 $50 \times (13 + 5 \times 3)$ |

Table 2: would it be possible to differentiate those numbers between soil and vegetation parameters? I'm curious to know for instance how root depth parameters are affected by DA.

Root depth parameters are not estimated in this DA problem, since they did not appear to be influential on moisture variables in a previous study as said in the section 2.4 DA setup : "a global sensitivity analysis of PESHMELBA in this case study showed that parameters that most influence moisture profiles are mainly θs (water contents at saturation) for the different soil horizons (Radišić et al., 2023)." => We changed all the Figure and Table captions to make it more clear.

**4.3 On the limitations of the methods**

 L444: From where those correlations are calculated/derived? is this somehow related to the state covariance matrix?

These are indeed the sampled correlation produced by the ensemble restricted to the parameter part.

 L444: As the state is not perturbated initially, I'm curious to know what the correlations look like at time 0; How about showing the correlation at time 0 to see the evolution with time (in appendix?).

The figure below shows correlations at any time of the simulations. These are the portions of the correlation matrices from the free runs corresponding to the surface moisture trajectory over the UH10. The black trajectories at the top and left of each matrix represent the mean trajectories of each ensemble, while the vertical and horizontal black lines indicate the times at which observations are available. We consider this figure a bit difficult to interpret and did not think this would be useful for the reader, this would represent a new discussion that would make the paper too long to our opinion. However, this is possible to add it in the appendix if you think this can help.

[Figure]

**Code availability**

 I appreciate seeing that the study can be reproducible with data accessible and open-source codes. Thanks to the authors for this effort. thanks!

**Appendix A**

 - Please explain the nomenclature in the Pdf column: what are N, TN, and LN (Normal, Log-Normal, …)

=> Done

- isnt a rooting depth of 0.9m as a nominal value too high for grassland? (Vegetation parameters for VFZ)

This value is based on a large study on VFZs over Europe from Brown et al. 2007:

Brown, C., A. Alix, J. L. Alonso-Prados, D. Auteri, J. J. Gril, R. Hiederer, C. Holmes, A. Huber, F. de Jong, M. Liess, S. Loutseti, N. Mackay, W. M. Maier, 218S. Maund, C. Pais, W. Reinert, M. Russell, T. Schad, R. Stadler, M. Streloke, M. Styczen et J. van de Zande (2007). Landscape and mitigation factors in aquatic risk assessment. Volume 2 : detailed technic. Rapp. tech. European Commission.

---

## Author Comment (AC2)

**RC2**: 'Comment on hess-2024-219', Anonymous Referee #2, 04 Oct 2024 reply

We thank very much the reviewer 2 for his positive comments on our study and took into account most of his comments, improving the paper.

Authors presented a comparison of different DA methods in the context of modular hydrological model for water quality management. The paper is well-written and looks like very comprehensive. I have a couple of comments:

1) In the literature, a few papers about the comparison of DA methods have been published in the field of hydrogeology. Also, those methods are well established, and the disadvantages and advantages are well-known.

This is true about the classical EnKF, less so when looking at iEnKS et ES-MDA which have seldom been used (if ever) for such application and behave very differently from EnKF. Moreover PESHMELBA has some significant peculiarities (see below) that make this study necessary.

Authors highlighted the modular hydrological model in this study instead of many studies using numerical models. if so, authors should clarify why there are differences using different physical models for DA, not only from the results of DA experiments, but from the methodology. Fundamentally, DA methods such as EnKF can be coupled with any transfer functions.

The PESHMELBA is said to be "modular" in the way that it is a coupling of several (independant) modules, each ones representing the processes occurring in a specific element of the landscape and playing a role in pesticide transfers : a vegetative filter strip, a river, a plot of maize, a hedge, etc etc. The meshing is this not a classical one but based on the landscape management leading to hydrological units playing a specific role in the agricultural catchment. Finally, the processes may be physically-based modeled when it is possible (for instance, Richards equation for infiltration in plots), or more conceptually/empirically when there is no equation known to represent it. The aim of this model is not to be a fully physically-based model such as Parflow or Hydrus-3D, but to simulate and compare scenarios of landscape management (e.g., including more or less buffer zones in a catchment), to identify an optimal configuration regarding pesticide transfer mitigation and demonstrate to the stakeholders. For these reasons, the model is a coupling of physical and empirical/conceptual models (also called "semi-conceptual models" in Buytaert et al. 2008) and can depend on thresholds making it highly non-linear. This type of model has not been widely investigated for data assimilation. We can imagine that it will be difficult to find a regular solution for these semi-conceptual models and this is why we think this study is important in data assimilation for hydrological and water quality modeling for decision-making.

This is explained in lines 66-77, and maybe also clarified by the new title : "Comparison of ensemble assimilation methods in a decision support model for landscape management to mitigate pesticide transfer"

Buytaert, W., Reusser, D., Krause, S., and J.-P., R.: Why can't we do better than Topmodel?, Hydrological Processes, 22, 4175–4179, https://doi.org/10.1002/hyp.7125, 2008.

2) in line 273, the true value comes from perturbation from Gaussian noises. does this mean that your ground truth has a Gaussian distribution. How is this close to the real data? Does the real data follow Gaussian distribution? If it has a non-Gaussian distribution, how does those DA methods perform?

This a very good question. Indeed, these methods all assume that the probability densities being manipulated are Gaussian in order to be optimal, which was verified in a previous work (Rouzies et al., 2023, and Rouzies PhD, 2024 (in french)). We added this in the text.

However, such an assumption is not justified in all cases of Rouzies PhD. In particular, it is noted that in some winter scenarios we tested, surface humidity sometimes follows a bimodal distribution. In these cases, a particle filter approach (van Leeuwen and Evensen, 1996) may be an interesting alternative, as it does not rely on any assumption of Gaussianity. This method has rarely been applied in hydrology (Moradkhani et al., 2005 ; Pasetto et al., 2012), although the particle filter remains an attractive method that may be worth exploring in cases where the ensemble Kalman filter fails.

A simpler solution to continue using the EnKF, consists in transforming variables into gaussian ones, using anamorphosis methods (Bertino et al., 2003). See, for instance, applications of anamorphosis in ocean and biogeochemical modeling (Beal et al., 2010) or in meteorological reanalysis (Devers et al., 2020).

Béal, D., Brasseur, P., Brankart, J.-M., Ourmières, Y., and Verron, J.: Characterization of mixing errors in a coupled physical biogeochemical model of the North Atlantic: implications for nonlinear estimation using Gaussian anamorphosis, Ocean Sci., 6, 247–262, https://doi.org/10.5194/os-6-247-2010, 2010.

Bertino, L., Evensen, G., and Wackernagel, H.: Sequential Data Assimilation Techniques in Oceanography, International Statistical Review, 71, 223–241, https://doi.org/10.1111/j.1751-5823.2003.tb00194.x, 2003.

Rouzies, E., Lauvernet, C., Sudret, B., and Vidard, A.: How to perform global sensitivity analysis of a catchment-scale, distributed pesticide transfer model? Application to the PESHMELBA model, Geoscientific Model Development, 2023, 1–44, https://doi.org/10.5194/gmd-16-3137-2023, 2023.

Rouzies, E. Quantification et réduction de l'incertitude dans un modèle de transfert de pesticides à l'échelle du bassin versant. Mathématiques [math]. Université Grenoble Alpes [2020-..], 2023. Français. ⟨NNT : 2023GRALM025⟩. ⟨tel-04659164v2⟩

3) As we know, those DA methods are impacted by the ensemble size.  Have you considered to implement some localizations to constrain the covariance so that the filter inbreeding issue could be reduced? In figure 11, it looks like that, if ensemble size is increased from 50 to 200, the performance of DA gets worse. This does not make sense.

Indeed, localization schemes could be implemented by using the covariance localization or local domain DA . About the covariance, this would be feasible with another implementation of the Kalman Filter, but not with ETKF since the covariance matrix is not built explicitly in this method. About the local domain DA, this would be very interesting and quite relevant, especially considering the structure of soils that are described (see Figure 12 that shows the spatial correlation by soil type).

In the discussion, l. 456, we added this sentence :

l 456. Figure 12 also highlights the absence of spatial correlations between soil units of different soils, advocating for using a scheme with local domain DA in ES-MDA (Asch et al., 2016) to alleviate the computational cost of this method that uses high dimension matrices.

Note that Figure 11 represents a test run. If reproduced multiple times, we would likely observe a trend of decreasing error. However, due to the high cost, we opted not to conduct further testing. The mean trend conforms to what we expected. We have performed this test for sizes 20 and 50, but extending it further is prohibitively expensive.

---

## Author Response (AR2)

Dr Claire Lauvernet
INRAE - Riverly
5 rue de la Doua
69625 Villeurbanne Cedex
France

Dear Editorial Board,

1. The ROR database lists the institution of the corresponding author but with a different city than given in the manuscript. Please clarify whether the ROR "Institut National de Recherche pour l'Agriculture, l'Alimentation et l'Environnement (Paris, France)" is still correct.

=> yes it is correct

2. For the next revision please use the initials instead of the full names of the authors for the section "Author`s contribution".

=> done

3. Some sections are misplaced (Code and data availability). Please pay attention to a manuscript composition. More information please see: https://www.hydrology-and-earth-system-sciences.net/submission.html#manuscriptcomposition

=> done

Best regards,

**Claire Lauvernet**